# Bacterial RadA is a DnaB-type helicase interacting with RecA to promote bidirectional D-loop extension

Léa Marie[1,*], Chiara Rapisarda[2,*,†], Violette Morales[1], Mathieu Bergé[1], Thomas Perry[2,†], Anne-Lise Soulet[1], Clémence Gruget[2], Han Remaut[3], Rémi Fronzes[2,†] & Patrice Polard[1]

Homologous recombination (HR) is a central process of genome biology driven by a conserved recombinase, which catalyses the pairing of single-stranded DNA (ssDNA) with double-stranded DNA to generate a D-loop intermediate. Bacterial RadA is a conserved HR effector acting with RecA recombinase to promote ssDNA integration. The mechanism of this RadA-mediated assistance to RecA is unknown. Here, we report functional and structural analyses of RadA from the human pathogen *Streptococcus pneumoniae*. RadA is found to facilitate RecA-driven ssDNA recombination over long genomic distances during natural transformation. RadA is revealed as a hexameric DnaB-type helicase, which interacts with RecA to promote orientated unwinding of branched DNA molecules mimicking D-loop boundaries. These findings support a model of DNA branch migration in HR, relying on RecA-mediated loading of RadA hexamers on each strand of the recipient dsDNA in the D-loop, from which they migrate divergently to facilitate incorporation of invading ssDNA.

[1] Laboratoire de Microbiologie et Génétique Moléculaires, UMR5100, Centre de Biologie Intégrative (CBI), Centre National de la Recherche Scientifique (CNRS), Université de Toulouse, UPS, Toulouse F-31062, France. [2] Department of Structural Biology and Chemistry, G5 Biologie structurale de la sécrétion bactérienne, UMR 3528, CNRS/Institut Pasteur, Institut Pasteur, 25-28 rue du Docteur Roux, Paris 75015, France. [3] Structural and Molecular Microbiology, Structural Biology Research Center, VIB, Pleinlaan 2, Brussels 1050, Belgium. * These authors contributed equally to this work. † Present address: Institut Européen de Chimie et Biologie, UMR 5234 Microbiologie fondamentale et pathogénicité CNRS/université de Bordeaux, 2 rue Robert Escarpit, 33607 Pessac, France. Correspondence and requests for materials should be addressed to R.F. (email: r.fronzes@iecb.u-bordeaux.fr) or to P.P. (email: patrice.polard@ibcg.biotoul.fr).

Bacteria are only able to generate clonal descendants, and their genetic diversity is maintained by the accumulation of random point mutations. As such, they should be extremely vulnerable to changes in their environment. Instead, these organisms are extremely adaptable and capable of adjusting their lifestyle very quickly when changes occur. One dramatic illustration of this capacity is the spread of antibiotic resistance among bacterial pathogens[1] due to antibiotics overuse in human healthcare and livestock[1]. The discovery of penicillin by Alexander Fleming in 1928 ushered in the antibiotic era. Nowadays, about 30 antibiotics represent our only line of defence against many bacterial pathogens[2]. During the last decade, the emergence of multi-resistant bacteria, refractive to several treatments, led to increased mortality caused by common infections[3].

In this context, it is crucial to fully understand the molecular mechanism of bacterial adaptability to ultimately target and limit it. Among the several processes involved, homologous recombination (HR) is key, since it is fundamental to the maintenance and the plasticity of bacterial genomes. This process is also a key step of horizontal gene transfer processes such as conjugation and natural transformation, facilitating the acquisition of new genetic traits. The different HR pathways are driven in their early steps by a common multitasking motor protein (referred herein to as an HR recombinase) highly conserved in the three domains of life, named RecA in bacteria, Rad51 and Dmc1 in eukaryotes and RadA in archaea[4,5]. They catalyse the DNA exchange reaction between complementary strands in an ordered and ATP-dependent manner, as depicted for RecA in Fig. 1a. RecA first assembles into a helical filament on single-stranded DNA (ssDNA), growing mainly in the 5′ to 3′ direction[5,6]. Next, the resulting nucleofilament identifies and pairs the bound ssDNA with homologous double-stranded DNA (dsDNA), and promotes DNA branch migration towards the 3′ end of the invading DNA strand[7]. These consecutive reactions driven by HR recombinases result in a triple-stranded DNA product, commonly named a 'Displacement-loop' (D-loop). Next, the D-loop is processed further and finally resolved to restore the DNA duplex.

Other types of motor proteins are also known to promote DNA branch migration reactions in distinct HR pathways. They all promote homologous DNA strand exchange on preformed branched DNA molecules. The bacterial RuvB and RecG DNA helicases are among the best characterized of these HR effectors[8]. They promote DNA migration at DNA junctions by a distinct mechanism relying on their ATP-dependent translocation along dsDNA. RuvB acts on two opposite dsDNA arms of four-way Holliday junctions (HJ), with the assistance of RuvA specifically bound at the crossing of the four DNA arms. Once assembled at the HJ, the RuvAB complex functions as a dual DNA pump energized by the ATP hydrolysis at the level of two RuvB hexameric rings encircling the two duplex arms of the HJ[9]. RecG acts as a monomer on various branched DNA substrates, including HJ, three-way junctions (such as those that define the boundaries of the HR D-loop) and forked DNA molecules. Bound to these branched DNA substrates, RecG promotes DNA branch migration by driving simultaneously dsDNA unwinding and rewinding of complementary DNA strands[10,11]. A third emerging HR effector able to promote DNA branch migration in bacteria is the RadA/Sms protein (named hereafter as RadA, and distinct from the archeal RadA HR recombinase). RadA has been historically identified and characterized in *Escherichia coli*, where it has been genetically defined as being involved in the recombinational repair of damaged DNA[12]. This was confirmed in *Bacillus subtilis* and *Streptococcus pneumoniae*, for which RadA was also found to be involved in the dedicated HR pathway of natural genetic transformation[13,14], an horizontal gene transfer

process that takes place during the particular physiological state of bacterial competence[15,16]. Further *in vivo* analysis of RadA function in HR pathways of genome maintenance in *E. coli* have highlighted its functional redundancy with RecG and RuvAB[12,17]. Recently, biochemical analysis of purified *E. coli* RadA showed that it acts at a 3-way junction (3-J) made by RecA to promote branch migration in an ATP-dependent manner, either with or without RecA[18]. In this DNA transaction reaction, RadA mimics the DNA strand exchange reaction catalysed by RecA in the 3′ direction of the ssDNA paired to the homologous dsDNA template. On the basis of the sequence homology shared between the central domain of RadA and the ATP binding core domain of RecA[19], it was therefore proposed that RadA would act like, and possibly with, RecA, through a coordinated binding of the invading ssDNA and recipient dsDNA[18].

To gain insight into the molecular mechanism of RadA function during HR, we combined an *in vivo* analysis of RadA function in genetic transformation in *S. pneumoniae* with its structural and biochemical characterization *in vitro*. We find that RadA is required to recombine ssDNA across long distances in the genome during transformation. This genetic study highlighted the key role of RadA in extending RecA-driven transformation D-loops either in the 3′ or 5′ direction relative to the polarity of the invading ssDNA. The crystal structure of pneumococcal RadA revealed that its central domain adopts the same fold as the helicase domain of bacterial replicative DnaB-type helicases. In addition, like DnaB proteins, we show that RadA assembles into a ring-shaped hexamer and is a functional DNA helicase that loads onto ssDNA and translocates in a 3′ direction along it to unwind dsDNA. As such, RadA unwinds the 5′ tailed dsDNA of synthetic 3-J substrates. We also found that RadA interacts with RecA, which allows unwinding of the 3′ tailed dsDNA of the 3-J substrate. Altogether, these findings support an unprecedented, to the best of our knowledge, model for a DNA branch migration mechanism in HR mediated by RadA. The helicase migrates divergently from the boundaries of the HR D-loop and is loaded by RecA on both strands of the recipient dsDNA.

## Results

**RadA promotes the extension of transformation D-loops.** Natural genetic transformation is driven by a dedicated multi-protein machinery, named the transformasome[20], which assembles in the cell membrane and promotes the binding of exogenous dsDNA on the surface of competent cells, the internalization of one DNA strand and its integration into genomic DNA via RecA-driven HR (Fig. 1b). In the human pathogen *S. pneumoniae*, all cells become competent for ∼20 min in response to various stresses, including some antibiotics[15,21–23]. Pneumococcal transformation frequency measured for a selective point mutation present within fully homologous chromosomal donor DNA is reduced 100-fold in cells lacking *radA*[13] (Supplementary Fig. 1a). One explanation of this deficit is that RadA may act to extend ssDNA incorporation at HR transformation D-loops, a proposal supported by the recent finding that *E. coli* RadA can promote DNA branch migration *in vitro*[18]. If so, RadA would maximize the amount of transforming ssDNA integrated within the genome, and thus the number of mutations acquired by competent pneumococci. To test this possibility, we repeated the transformation experiments by using donor PCR fragments of similar size (∼4 kbp) carrying a selectable point mutation located either centrally (PCRc) or close to one end (PCRe; Fig. 1c). In contrast to chromosomal donor DNA, the use of PCR donor fragments raises the amount of transforming ssDNA molecules internalized per cell in the population (Fig. 1d and Supplementary Fig. 1b).

Transformation frequency is proportional to the amount of PCRc donor DNA added and can exceed 85% (Supplementary Fig. 1c). Random cleavage of these PCR fragments, followed by ssDNA degradation and internalization by the 3′ end (Fig. 1b), alters the position of the point mutation on the transforming ssDNA molecules (Fig. 1d). For PCRc, the mutation will be equally distributed between the middle and the 3′ end of the transforming ssDNA molecules. For PCRe, the mutation will be located almost exclusively near the 5′ end. The *radA* mutant exhibited a frequency of transformation with PCRc nearly 50-fold lower than

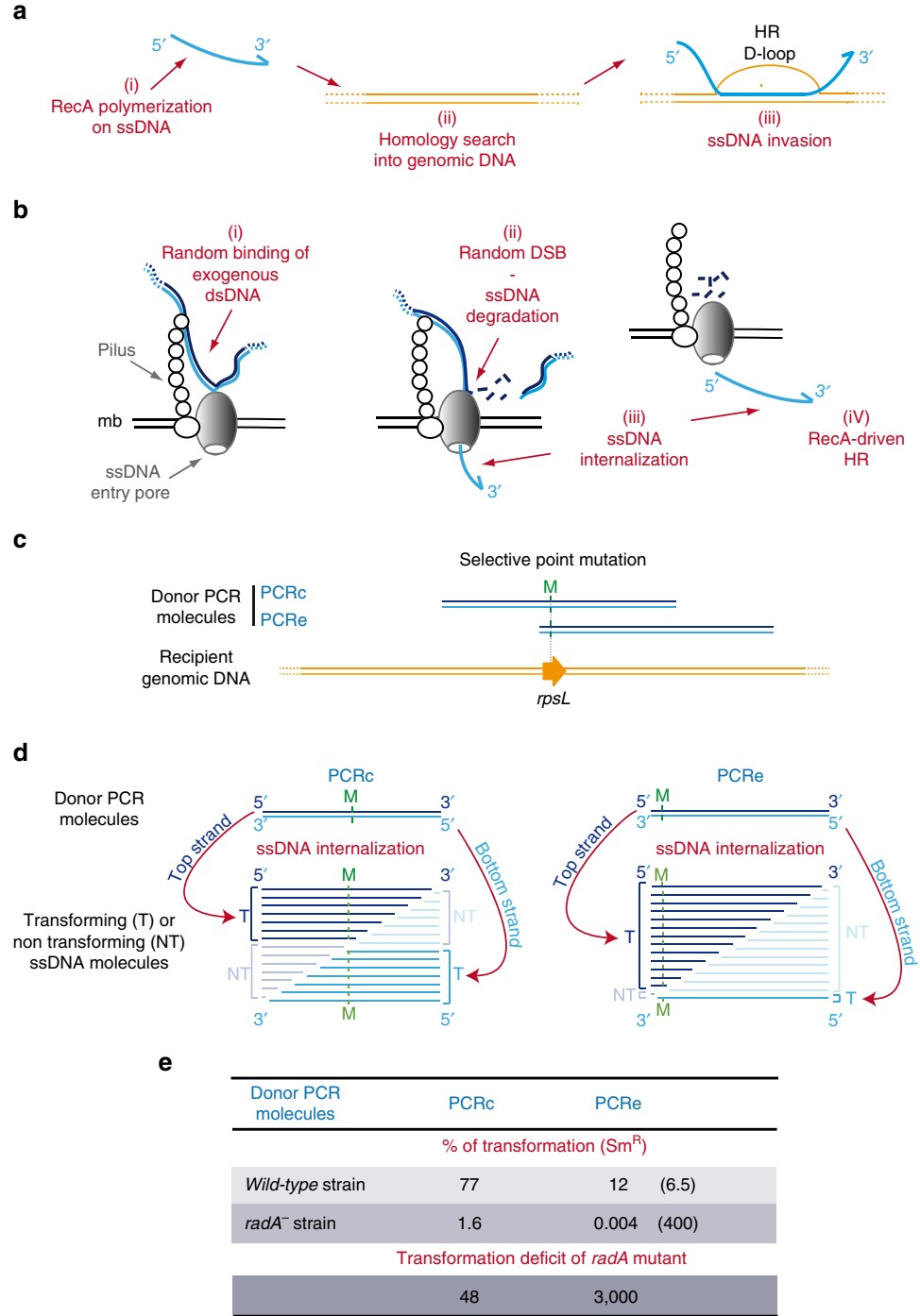

**Figure 1 | RadA promotes HR extension during pneumococcal genetic transformation.** (**a**) Schematic representation of the three main early steps of RecA-driven homologous recombination (HR). (**b**) Schematic representation of the three main steps of pneumococcal genetic transformation. (**c**) Donor and recipient molecules of the streptomycin resistance (SmR) in the *rpsL* gene in the transformation assay. (**d**) Schematic representation of ssDNA molecules created after random cleavage of the DNA imported into competent cells from either donor strand of PCRc (left) or PCRe (right). Transforming ssDNA fragments (T; thick lines) carry the point mutation (M), a mutation into the *rpsL* gene (SmR); Non-transforming ssDNA fragments (NT; thin lines) lack the point mutation and account for half of ssDNA molecules internalized. (**e**) Frequency of transformation of wild-type and *radA*⁻ strains by PCRc and PCRe donor DNA (20 ng ml⁻¹). The number in parenthesis indicates relative deficit of transformation between PCRc and PCRe donors in the same strain. Data are from three independent experiments. See also Supplementary Fig. 1 for transformation assays performed with donor PCR fragments harbouring different selective point mutations, which produced similar results.

wild-type, at a ratio of DNA molecules per cell estimated at ∼100. Remarkably, in the same conditions, transformation of the *radA* mutant with PCRe was 3,000-fold less frequent than wild type (Fig. 1e). Similar results were obtained with two other selectable point mutations (Supplementary Fig. 1d). In addition, the use of large amounts of PCRc donor DNA diminished the need for RadA in transformation (Supplementary Fig. 1c), suggesting that RadA acts to potentiate RecA-directed incorporation of transforming ssDNA in the genome. The lower transformation frequency measured in *radA*⁻ cells transformed with PCRe compared to PCRc strongly supports the notion that RadA extends ssDNA incorporation from RecA-driven transformation D-loops. Importantly, the strong transformation deficit obtained with the PCRe donor fragments also indicate that RadA could promote DNA branch migration in the 5′ direction of the D-loop, thereby improving the integration of transforming ssDNA molecules over extensive genomic distances.

**Crystal structure of pneumococcal RadA**. All bacterial RadA proteins are characterized at the sequence level by three conserved domains[12] (Fig. 2a): a potential C4-type zinc binding motif (C4); a central canonical RecA-like ATPase domain (H), including a KNRFG motif unique to and strictly conserved in RadA proteins; and a region homologous to the protease domain (P) of the bacterial Lon protein. The full-length protein (RadA$_{FL}$) and the P domain (RadA$_P$) of pneumococcal RadA were purified from *E. coli* and shown to assemble predominantly as hexamers in solution (Supplementary Fig. 2a–c). The assembly into hexamers is neither dependent on nucleotide binding nor on the presence of $Mg^{2+}$ ions. The crystal structure of RadA$_P$ was solved at 2.5 Å resolution (Table 1, Supplementary Fig. 10) and the structure of RadA$_{FL}$ bound to thymidine diphosphate (dTDP) was solved at 3.5 Å resolution (Table 1, Supplementary Fig. 11). The structure of RadA$_{FL}$ reveals three protomers per asymmetric unit, organized as annular hexameric particles with two-fold symmetry (Fig. 2b, Supplementary Fig. 2a,b) and forming a two-tiered planar ring, corresponding to the H and P domains (Fig. 2a). The C4 domain (residues 1–35), present but unresolved in the crystal structure, likely occupies the gap observed in the crystal packing and is a functional zinc finger (Supplementary Fig. 2d,f). Indeed, RadA$_{FL}$ binds zinc and to a smaller extent iron (Supplementary Fig. 2d,f), while a RadA derivative deleted of the C4 domain (RadA$_{\Delta C4}$) was no longer able to bind the divalent metals (Supplementary Fig. 2e,g).

The central H domain and the clasp in the RadA$_{FL}$ structure (residues 53–274) forms a ring with two-fold symmetry, with outer and inner diameters of 100 and 25 Å, respectively (Fig. 2b). It displays a typical RecA-fold containing a nucleotide-binding site (Fig. 2c) and it is, most likely, the functional core. The position and dynamics of the H domain are different in each protomer of the asymmetric unit regardless of the nucleotide bound (Supplementary Fig. 3a,b). In the RadA$_{FL}$ structure obtained upon crystallization in the presence of dTTP, each protomer was occupied by a hydrolysed dTDP. The nucleotide binding site displays all the canonical residues found in an ATPase active site of the RecA ATPase family[24] (Fig. 2c). However, no arginine finger, required for nucleotide hydrolysis was found close to this site.

While RadA is considered to be a RecA homologue on the basis of the sequence of its central H domain[18], fold recognition servers such as Dali[25] or PDBeFold[26] revealed that its crystal structure is similar to that of the helicase domain found in ring-shaped hexameric replicative helicases of the DnaB family, with Gp4 of bacteriophage T7 (ref. 27) being its closest structural paralogue (Fig. 3a). DnaB interaction with ssDNA involves two loops, L1

and L2, located inside the hexameric ring[28], which are found unresolved at the same location in the RadA$_{FL}$ crystal structure (Figs 2a and 3a). The H domains from both Gp4 and RadA share positively charged external surfaces and a clasp that links one domain to the other (Fig. 3b,c). The structural similarity between the two proteins is limited to the H domain, which is encompassed by different domains (Fig. 3d,e).

In the structures of RadA$_P$ and RadA$_{FL}$, the P domain (residues 290–453) forms a ring with six-fold symmetry (Figs 2b and 3g). In RadA$_{FL}$, the outer and inner diameters of the P ring are 100 and 15 Å, respectively (Figs 2b and 3g). RadA$_P$ is structurally related to the proteolytic domain of Lon proteases (Lon$_P$), which also assembles into a stable annular hexamer, within the full-length protein and as an independent domain (Fig. 3f,g)[29]. However, the typical lysine and serine residues of the protease catalytic dyad in Lon$_P$ are replaced in the pneumococcal RadA$_P$ domain by an arginine and an alanine, respectively (Figs 2c and 3f). Furthermore, even though the sequence and the fold are similar, the electrostatic potentials of the inner side of the rings have opposite charges, namely negative in the Lon$_P$ and positive in RadA$_P$ (Fig. 3g,h). Thus, RadA$_P$ appears to be structurally but not functionally related to LonP, both adopting the same ring-shaped hexameric architecture. In both RadA$_{FL}$ and RadA$_P$, the hexameric structure is primarily maintained by the extensive surface of interaction between adjacent P domains (940–976 Å). At the level of the H domain, inter-protomer contacts (268–848 Å) are enhanced by the N-terminal clasp (residues 53–65) that winds around the neighbouring H domain between two H domains (Figs 2c and 3b). Importantly, most of the residues conserved among RadA proteins map at the interfaces between the monomers and inside the H and P rings (Supplementary Fig. 7c), indicating that this hexameric ring should be a conserved feature of RadA proteins.

**RadA is a helicase involved in homologous recombination**. As mentioned above the H domain of RadA is a close structural paralogue of the helicase domain of Gp4, a prototypical member of the DnaB subfamily of SF4 helicases. These mostly hexameric proteins unwind dsDNA by encircling a ssDNA strand and translocating along this strand exclusively in the 5′ to 3′ direction, a process fuelled by NTP hydrolysis (generally ATP[27]). To test whether the structural similarity to SF4 helicases was also functional, we investigated the ATPase and dsDNA unwinding activities of RadA. We recombinantly expressed and purified full-length native pneumococcal RadA, together with four RadA derivatives containing alanine substitutions for key residues of the H and C4 domains (Supplementary Fig. 4a,b): K101A in the Walker A motif of the ATPase domain, K251A and R253A in the KNRFG motif (Fig. 4a) and C27A in the C4 domain. All four mutant proteins assemble mainly as hexamers in solution, except for the RadA$_{C27A}$ mutant that forms a mixture of multimers and hexamers (Supplementary Fig. 4b). Wild-type RadA displays hardly any basal ATPase activity. This activity is stimulated 10-fold by dsDNA and 50-fold by ssDNA (Fig. 4b and Supplementary Fig. 4c). Notably, stimulation of RadA$_{K101A}$ ATPase activity is severely reduced in the presence of ssDNA (Supplementary Fig. 4c).

Next, we showed that RadA is an active helicase, by using agarose gel electrophoresis to detect unwinding of a fluorescent oligonucleotide hybridized to a large circular ssDNA molecule (Fig. 4c). RadA was found unable to unwind a full dsDNA linear substrate (Fig. 4d), indicating that initiation of helicase activity requires a ssDNA overhang. Indeed, using short 3′ or 5′ tailed duplexes, we found that RadA unwinds dsDNA by translocating along ssDNA from 5′ to 3′ (Fig. 4e), a feature common to DnaB helicases[24,27]. RadA helicase activity is as efficient on forked DNA

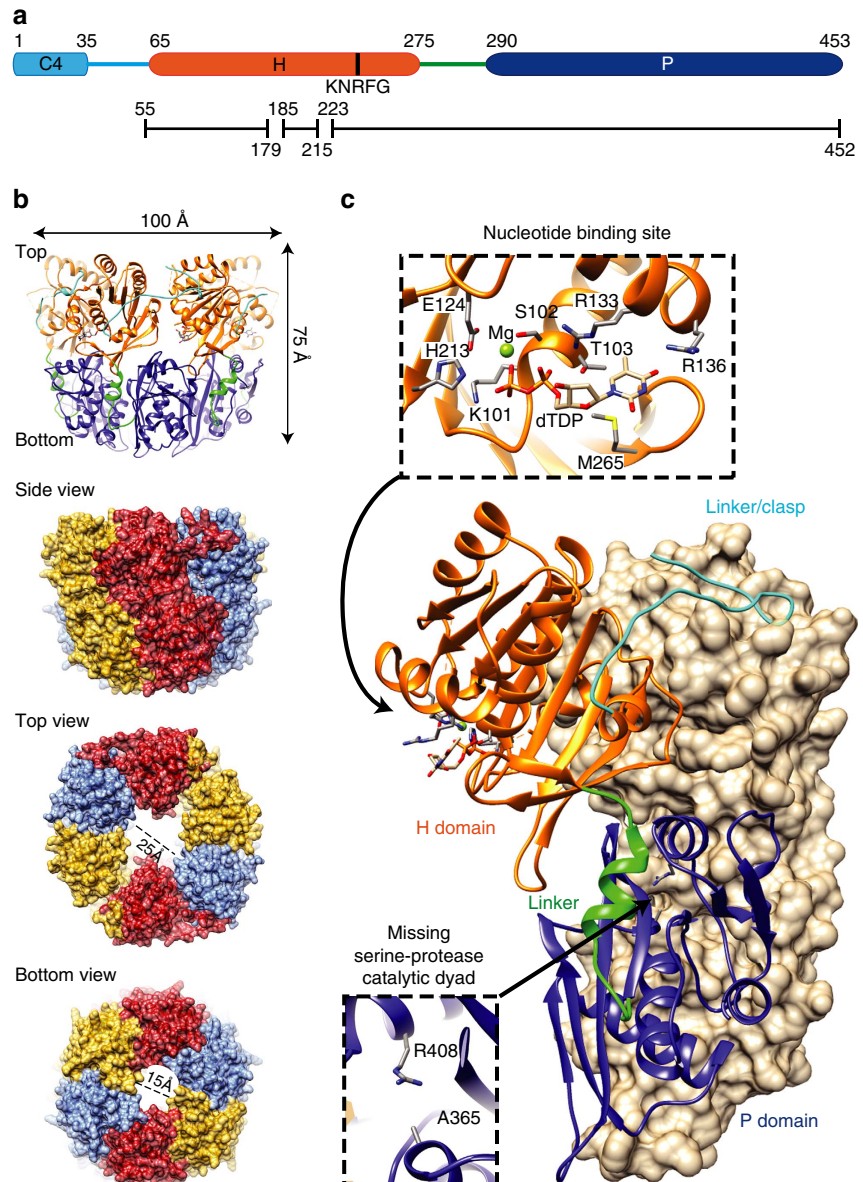

**Figure 2 | Crystal structure of pneumococcal RadA.** (**a**) Domain organization common to all bacterial RadAs. The numbering above the C4, H and P domains represents pneumococcal RadA amino acids (453 residues long). The black lines denote structurally determined regions in this study, solved from crystals of RadA$_{FL}$ and RadA$_{P}$. (**b**) Structure of pneumococcal RadA determined from a crystal of dTDP-bound RadA$_{FL}$. The C4 domain is present but not structured in the crystal. The same colour code as in panel A has been used for the H and P domains, and the two linkers flanking the H domain. The RadA trimer in the asymmetric unit forms planar and annular hexamers 100 Å wide and 75 Å tall. The internal diameter of the ring is 25 Å at the H domain and 15 Å at the P domain. See Supplementary Figs 2 and 3. (**c**) The oligomerization interface shows that contacts between interacting protomers occur throughout the H and P domains, with the N-terminal loop (in light blue) strengthening the contact. The nucleotide binding site in the H domain contains the canonical Walker A and Walker B motifs, with M265 stabilizing the base of the dTDP bound in each protomer in the crystals. The lysine and serine residues defining the active proteolytic site of the P domain of *E. coli* Lon protease correspond to an arginine (R408) and an alanine (A365) in the P domain of pneumococcal RadA. See also Supplementary Fig. 2.

as on tailed DNA (Fig. 4e), contrasting with DnaB that restrictively unwinds forked dsDNA encountered during its translocation along ssDNA[30]. In agreement with their deficiencies in ATP hydrolysis, the helicase activity was not observed with the K101A, K251A and R253A mutants (Fig. 4c). Interestingly, the RadA$_{C27A}$ mutant displays both ATPase and helicase activities, implying that the integrity of the C4 domain is not needed for ATP hydrolysis or to unwind dsDNA (Fig. 4b,c and Supplementary Fig. 4c).

Modelling the H domain of RadA on that of the previously solved structure of DnaB from *Geobacillus stearothermophilus*

bound to ssDNA, identifies K251 and R253 of the conserved KNRFG motif as respectively, the potential 'piston' and the 'arginine finger' of the ATPase active site of the neighbouring protomer[31]. Consistent with this proposal, RadA$_{K251A}$ and RadA$_{R253A}$ mutants displayed lower ssDNA-induced ATPase and helicase activities than wild-type RadA (Fig. 4b,c and Supplementary Fig. 4c). In the RadA crystal structure, R253 is located further away from the ATPase active site of the neighbouring protomer (11 Å) than the corresponding residue in DnaB helicases[31]. This difference could explain the very low basal ATPase activity of RadA.

**Table 1 | Crystallography data collection and refinement statistics.**

|  | RadA$_P$ SeMet | RadA$_{FL}$ SeMet |
|---|---|---|
| *Data collection* |  |  |
| Space group | R 3:H | I 2 2 2 |
| Cell dimensions |  |  |
| a, b, c (Å) | 88.5, 88.5, 127.5 | 99.4, 169.9, 187.4 |
| α, β, γ (deg) | 90, 90, 120 | 90, 90, 90 |
| Wavelength | 0.9792 | 0.9792 |
| Resolution (Å) | 44.23 − 2.498 (2.588 − 2.498) | 49.21 − 3.5 (3.625 − 3.5) |
| R-merge | 0.1295 (0.9128) | 0.1837 (1.071) |
| R-meas | 0.1427 (1.006) | 0.1852 (1.079) |
| R-pim | 0.05944 (0.4208) | 0.02295 (0.1316) |
| Mean I/sigma(I) | 10.91 (1.60) | 24.73 (6.59) |
| Completeness (%) | 1.00 (0.99) | 1.00 (1.00) |
| Multiplicity | 5.7 (5.6) | 66.5 (66.8) |
| CC$_{1/2}$ | 0.996 (0.629) | 0.998 (0.964) |
| *Refinement* |  |  |
| Unique reflections | 12909 (1296) | 20413 (2010) |
| R-work | 0.1860 (0.2544) | 0.1854 (0.2222) |
| R-free | 0.2449 (0.3536) | 0.2315 (0.2625) |
| Number of non-hydrogen atoms |  |  |
| Protein | 2,680 | 8,530 |
| Ligands | 102 | 78 |
| B-factors |  |  |
| Protein | 55.38 | 149.12 |
| Ligands | 53.38 | 203.83 |
| R.m.s.d |  |  |
| Bond lengths (Å) | 0.006 | 0.021 |
| Bond angles (deg) | 1.03 | 1.80 |
| Molprobity score | 1.49 | 2.14 |

Statistics for the highest-resolution shell are shown in parentheses.

The analysis of RadA interaction with long circular ssDNA molecules by electron microscopy (EM) revealed a 'pearl necklace' pattern of RadA hexamers, which appear arranged on their side (Fig. 4f). In the absence of DNA, RadA displays random orientations (Supplementary Fig. 2a). A similar pattern was previously reported for the Gp4 helicase by negative stain EM and interpreted as reflecting ssDNA encircled by hexameric rings[32]. Hence, the 'pearl necklace' pattern is specific to the interaction between RadA and ssDNA, most probably resulting from the encircling of the hexamer around the ssDNA template.

In electrophoretic mobility shift assays (EMSAs), RadA was seen to interact stably with short fluorescent linear ssDNA molecules, as well as dsDNA molecules, with a slight preference for ssDNA and with higher affinity in the absence of ATP (Supplementary Fig. 5a). Addition of competitor ssDNA to a preformed nucleo-protein complex displaced pre-bound RadA, demonstrating the dynamic nature of the RadA–ssDNA interaction (Supplementary Fig. 5b). The RadA$_{C27A}$ and RadA$_{K101A}$ mutants exhibited DNA binding activity similar to that of wild type, whereas the RadA$_{K251A}$ and RadA$_{R253A}$ mutants bound only minimally to ssDNA (Supplementary Fig. 5c,d). These results suggest that stable binding of RadA to DNA relies on an ATP-independent protein conformation. ATP binding and hydrolysis appears to switch RadA to another conformation, relocalizing the KNRFG motif for ATP hydrolysis and resulting in either efficient translocation of RadA along ssDNA or RadA release from dsDNA. Hence, in addition to defining the arginine finger of the ATPase domain shared between two adjacent H protomers, the KNRFG motif may act as a DNA sensor motif, or be structurally linked to it.

The expression levels of the strains possessing the four *radA* point mutants (RadA$_{C27A}$, RadA$_{K101A}$, RadA$_{K251A}$ or RadA$_{R253A}$) were comparable to wild-type RadA expression levels in competent cells (Supplementary Fig. 1e). All, but the RadA$_{K101A}$ mutant, exhibited a reduced transformation frequency, identical to the *radA*$^-$ strain. In the RadA$_{K101A}$ mutant, GT was ∼5-fold higher level than a *radA*$^-$ mutant and ∼20-fold lower than the wild-type strain (Supplementary Fig. 1a). Interestingly, the RadA$_{K101A}$ mutant was found to exhibit low ssDNA-dependent ATPase activity *in vitro* (Fig. 4b and Supplementary Fig. 4c). Altogether, these findings highlight the importance of the ATPase activity of RadA for HR during GT. Importantly, they also point to a key role of the C4 domain of RadA in GT, distinct from the ATPase and helicase functions of RadA.

**Interplay between RecA and RadA at three-way DNA junctions**. The finding that RadA is a hexameric helicase raises the question of the mechanism through which RadA promotes the extension of ssDNA incorporation during HR. To this end, we investigated the interaction of RadA with RecA and with branched DNA substrates that reflect the RecA-driven DNA strand exchange products.

We confirmed by a bacterial two-hybrid (BacTH) assay that RecA and RadA oligomerize (Supplementary Fig. 6a) and that they physically interact with each other (Fig. 5a). The RadA$_{C27A}$ mutant exhibits an attenuated interaction with RecA (Supplementary Fig. 6b) that could lead to the GT defect observed (Supplementary Fig. 1a).

In parallel, we studied the helicase activity of RadA on 3-way DNA junctions (3-J) mimicking those at the boundaries of the RecA-driven D-loop (Fig. 5b). We assembled two synthetic 3-J substrates, one with a 5′ ssDNA arm of 30 nt and a 3′ ssDNA arm of 70 nt (3-J1), and one with a 5′ ssDNA arm of 70 nt and a

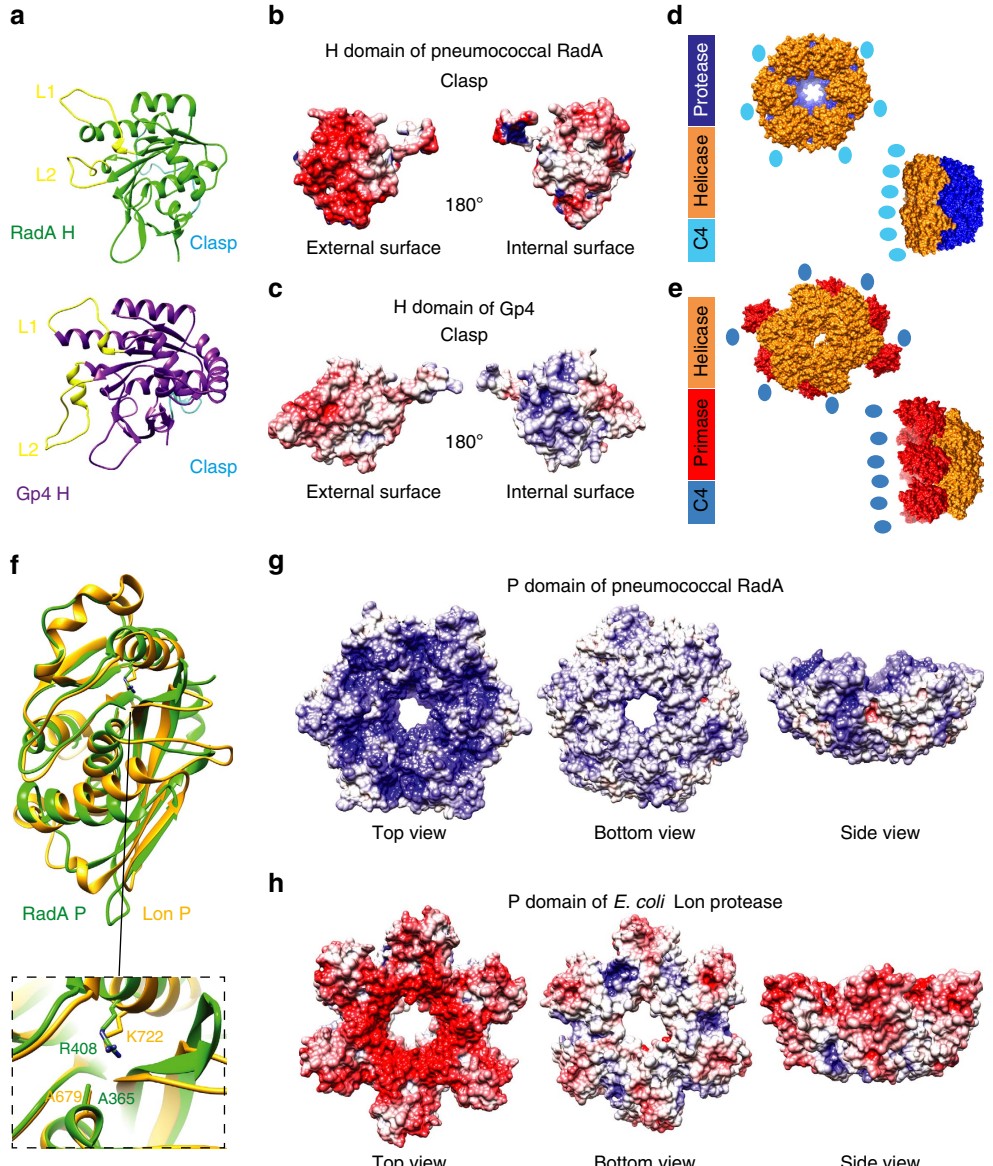

**Figure 3 | Comparison of RadA H and P domains with their structural paralogues.** (**a**) The RadA H domain is in green and the H domain of Gp4 T7 helicase/primase (PDB: 1CR4) in purple. The N-terminal loop acts as an interaction clasp with the adjacent protomer in both hexameric proteins is in light blue. The modelled L1 and L2 loops, demonstrated for some DnaB members to interact with ssDNA[28], are located inside the channel of the hexameric ring and found unstructured both in Gp4 T7 and in RadA crystals, are in yellow. The RMSD between the two structures is 0.940 Å for 118 atom pairs out of 191 atoms. (**b,c**) Electrostatic surface potentials of the H domain of RadA (**b**) and of Gp4 T7 (**c**) at pH 7 as calculated by the APBS server. (**d,e**) Comparison of the linear domain organization and quaternary structures of RadA (**d**) and the Gp4 T7 helicase-primase hybrid structure from PDBs 1E0J and 1Q57 (**e**). (**f**) The RadA$_P$ domain is in green and the *E. coli* Lon$_P$ domain in orange (PDB:1RRE). The folds of RadA$_P$ and Lon$_P$ are globally similar, but only 92 out of 177 atoms of the Cα chain match, with an RMSD of 1.050 Å. The inset shows the proteolytic active site of Lon$_P$ compared to the corresponding region in RadA$_P$: an arginine residue (R408) in RadA$_P$ is found at the location of the catalytic K722 of Lon$_P$; to obtain the Lon crystal structure, the catalytic S679 was mutated to Ala, a mutation inactivating the proteolytic activity of Lon. The corresponding residue in pneumococcal RadA$_P$ appears to be an alanine (A365). (**g,h**) Electrostatic surface potentials of RadA$_P$ (**e**) and *E. coli* Lon$_P$ (**f**), calculated with APBS software at pH 7. Blue: positively charged residues; red: negatively charged residues. The RadA$_P$ hexamer has a positively charged lumen, opposite to the DNA phosphate backbone that is presumed to translocate within this channel. Conversely, the Lon$_P$ domain is negatively charged and is mostly involved in protein folding during translocation within this channel.

3′ ssDNA arm of 30 nt (3-J2; Fig. 5c). One strand of these 3-J substrates was fluorescently labelled, to enable tracing of their unwinding by acrylamide gel electrophoresis (Fig. 5c and Supplementary Fig. 6d). The labelled S1 molecule is the main product generated from 3-J1 (Fig. 5d), while the F2b molecule is the main product from 3-J2 (Fig. 5e), in a RadA dose-dependent manner. Thus, RadA preferentially unwinds the 5′ ssDNA arm of both 3-J substrates. The level of unwinding of the 3-J substrates in

the '3′ direction' (named considering the polarity of the invading strand on the D-loop; Fig. 5b) is higher for 3-J2 than for 3-J1. This is possibly because the length of the dsDNA arm to unwind is shorter for 3-J2 (30 versus 70 bp), or because its longer 5′ ssDNA arm favours initial RadA loading. Helicase assays comparing unwinding efficiency of RadA targeting a 30 bp and a 70 bp duplex associated to a large circular ssDNA molecule strongly support the latter possibility (Supplementary Fig. 4d).

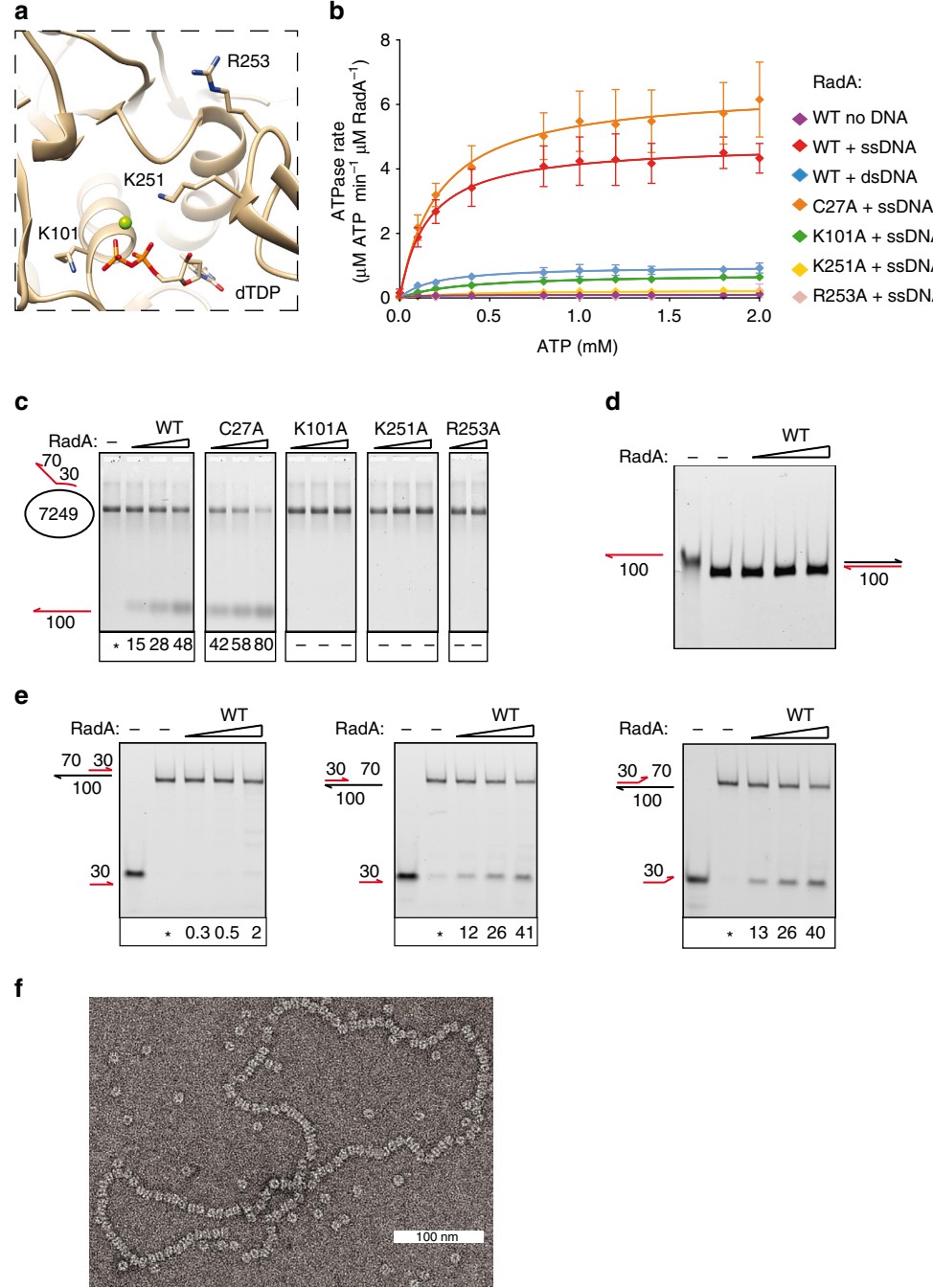

**Figure 4 | Pneumococcal RadA functions as an SF4 helicase.** (**a**) Overview of some of the residues in the ATPase site that were mutated in the functional study of RadA. (**b**) ATPase activity of RadA derivatives in the presence or absence of ssDNA or dsDNA, as indicated. RadA$_{R253A}$ data is concealed by the RadA wt data (red diamonds). See also Supplementary Figs 4 and 5. (**c–e**) Helicase assays performed with RadA derivatives and various DNA substrates (see text for details). The fluorescently (Cy3) labelled oligonucleotides are depicted in red. Numbers are the sizes of the DNA strands, in nt. Wedges show protein concentrations (in nM). Values of the relative amounts of released Cy3-DNA products (in % of total DNA) in each assay are reported below the images of the gels; the * indicates the assay without protein, from which the amount of unwound substrate has been quantified and retracted from the assays performed in the presence of protein (as detailed in Methods). (**f**) Negatively stained EM image of RadA in the presence of circular ssDNA and ATPγS.

At the highest RadA concentrations, the labelled S2 molecule is also observed, which could result from F2b unwinding. In addition, a very low-level of unwinding in the opposite '5′ direction' (Fig. 5b) was detected for both substrates resulting in F1a from 3-J1 and F2a from 3-J2 (Fig. 5d,e).

We next reproduced the helicase assays performed at the highest RadA concentration in the presence of purified pneumococcal RecA (Fig. 5f,g). Remarkably, RecA promoted unwinding of the 3-J substrates in the 5′ direction. This result suggested that RecA facilitates RadA loading on the bottom ssDNA strand at the junction of the 3-J molecules, thereby orienting RadA-driven ssDNA translocation and dsDNA unwinding in the 5′ direction. Notably, at the highest RecA concentration tested on 3-J1, the level of unwinding in the

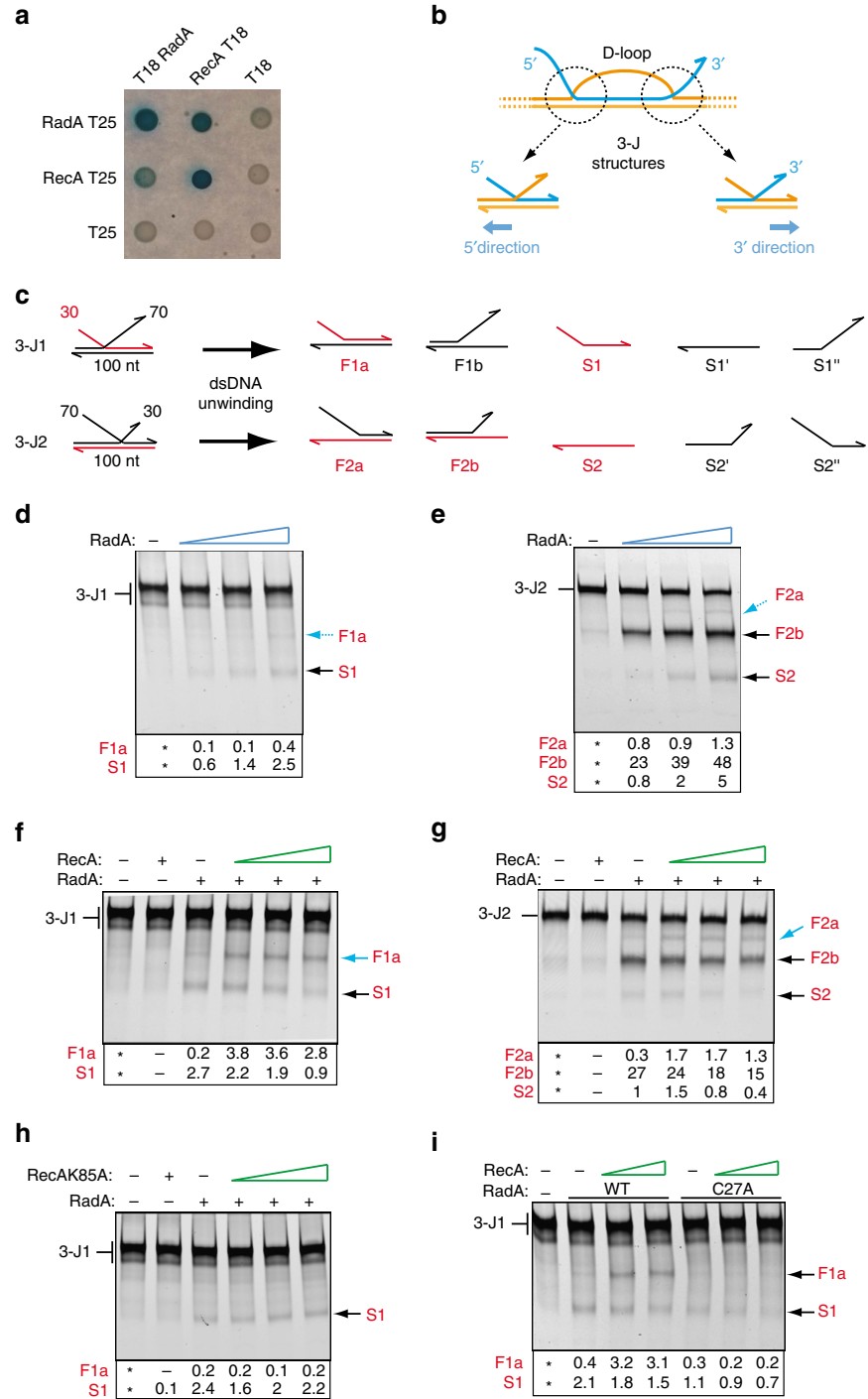

**Figure 5 | Interplay between RadA and RecA.** (**a**) RadA and RecA self- and cross-interact with each other in bacterial two-hybrid assay. See also Supplementary Fig. 6. (**b**) Schematic representation of the HR D-loop highlighting the two 3-J structures that define its boundaries. 5′ and 3′ dsDNA unwinding directions of the recipient dsDNA are given according to the 5′ to 3′ polarity of the invading ssDNA (depicted in blue), respectively. (**c**) Synthetic 3-J1 and 3-J2 substrates used to analyse RadA helicase activity, and the products that could result from their unwinding. Products detected by fluorescence are depicted and labelled in red (F, forked DNA; S, ssDNA). See also Supplementary Figs 6 and 8. (**d,e**) Helicase assays performed on 3-J1 and 3-J2, respectively, with increasing amounts of RadA (80, 250 and 750 nM). Black and blue arrows indicate major and minor products, respectively. (**f,g**) Helicase assays performed on 3-J1 and 3-J2, respectively, with RadA (750 nM) in the presence of increasing amounts of pneumococcal RecA (300, 600 and 900 nM). Bold blue arrows indicate increases in products obtained at low level in experiments performed with RadA alone (in **d**-**e**). (**h**) Helicase assays on 3-J1 performed with RadA (750 nM) in the presence of the RecA$_{K85A}$ ATPase mutant (150, 300 and 600 nM). (**i**) Comparison of unwinding activities of RadA and RadA$_{C27A}$ (750 nM) on 3-J1 in the presence of RecA (300, 600 nM). See Supplementary Fig. 6. In panels **d** to **i**, values of the relative amounts of DNA products (% of total DNA) in each assay are reported below the images of the gels; the * indicates the assay without protein, from which the amount of unwound substrate has been quantified and retracted from the assays performed in the presence of protein (as detailed in Methods).

5′ direction was greater than in the 3′ direction (Fig. 5f). In addition, 3′ direction unwinding was reduced relative to that seen with RadA alone (Fig. 5f,g). This indicates that RecA impeded RadA action on the 5′ ssDNA strand, possibly by competitive binding to this strand. Strongly supporting this proposal, helicase activity of RadA on a 5′ tailed duplex was gradually decreased by increasing amounts of RecA (Supplementary Fig. 6e, right panel). The same experiment reproduced on a reciprocal 3′ tailed duplex did not lead to its unwinding (Supplementary Fig. 6e), showing that RecA did not act on RadA to reverse its translocation and unwinding polarity. Therefore, this experiment further supports the notion that RecA promotes RadA loading on the bottom strand of the 3-J substrates to promote their unwinding in the 5′ direction.

Interestingly, RecA-mediated and RadA-dependent unwinding of the 3-J1 substrate in the 5′ direction was found to depend not only on the ATPase activity of RadA (Supplementary Fig. 6f), but also on that of RecA, as shown using the ATPase domain mutant RecA$_{K85A}$ (Fig. 5h) that still interacts with RadA in the BacTH assay (Supplementary Fig. 6b). This suggests that RecA polymerizes on ssDNA to assist the loading of RadA on the bottom strand of the 3-J substrate. Finally, we tested the RadA$_{C27A}$ mutant, which is hindered in its interaction with RecA (Supplementary Fig. 6c) but retains wild type levels of ATPase and helicase activity (Fig. 4b,c and Supplementary Fig. 4c). Importantly, this mutant unwound 3-J1 in the 3′ but not in the 5′ direction, even in the presence of RecA (Fig. 5i).

These results demonstrate that RadA alone unwinds synthetic 3-J molecules mainly in the 3′ direction, but depends on the interaction with RecA for 5′ direction unwinding. The C4 domain is crucial for this functional interaction of RadA with RecA. Placing this RadA-dependent remodelling of 3-J structures at the boundaries of a RecA-created D-loop would result in unwinding of the parental dsDNA in both directions, the primary step of DNA branch migration.

## Discussion

We report here a detailed structural and functional analysis of the RadA protein from *S. pneumoniae*, which uncovers several key properties of this HR effector encoded in nearly all bacterial species and in some eukaryotes (Supplementary Fig. 7a). First, purified pneumococcal RadA self-assembles into an annular hexamer in solution. Second, its central RecA-like domain is a close structural paralogue of the helicase domain of DnaB-related SF4 helicases, which form a well-defined family of ring-shaped hexameric motors of bacterial replication forks, all unwinding dsDNA by translocating around the lagging ssDNA template.

Third, RadA is an active helicase, sharing several mechanistic similarities with DnaB. Both RadA and DnaB induce their ATPase activity in the presence of ssDNA and, fuelled by ATP hydrolysis, they catalyse dsDNA unwinding of forked DNA molecules in the 5′ to 3′ direction. On synthetic three-way DNA junctions (3-J) that mimic the boundaries of a D-loop, RadA has been found to unwind the 5′ tailed moiety of the branched DNA molecules. This reaction reflects the unwinding of the parental duplex DNA that should occur on one side of an HR D-loop, which corresponds to the 3′ direction of the invading homologous ssDNA. Fourth, we found that RadA interacts with RecA, and that this interaction promotes the unwinding of the second moiety of 3-J molecules, corresponding on a D-loop to the 5′ direction of the invading homologous ssDNA. These results imply that RadA can promote DNA branch migration of RecA-driven HR D-loops in both directions. They also explain the crucial requirement for RadA in the HR pathway of genetic transformation to promote the efficient incorporation of point mutations.

Bacterial RadA is a novel member of the superfamily 4 (SF4), thus, SF4 helicases are not restrictively involved in DNA replication but could also be specialized for acting in HR.

Nevertheless, RadA differs from DnaB helicases at several levels. The helicase domain of DnaB proteins is characterized by five canonical motifs, which are not all strictly conserved in RadA (Supplementary Fig. 7b). Conversely, RadA proteins present a functionally important KNRFG signature, which is not found in other SF4 members[18].

A feature that also markedly distinguishes RadA from other SF4 proteins is the presence of an additional and conserved C-terminal domain adjacent to the helicase domain, structurally similar to the protease domain of Lon proteases. The structure of pneumococcal RadA shows that its P domain forms a stable scaffold to assemble the helicase central motor in the full-length hexamer. By referring to the known orientation of the DnaB motor encircling ssDNA, we infer that the RadA P domain moves at the front of the hexamer in the 5′ to 3′ direction, at the branched DNA junctions of the D-loop to promote dsDNA unwinding. The internal diameter of the P domain (15 Å) can accommodate ssDNA but not dsDNA. This would provide RadA the capability to unwind 5′ tailed dsDNA as efficiently as forked DNA (Fig. 4e), by contrast to DnaB that is restrictively active on forked DNA templates[30]. In addition, the stability provided by the robust self-association of RadA P domain into a hexameric ring allows RadA to promote incorporation into the genome of transforming ssDNA over long distances.

Finally, another difference between RadA and DnaB proteins is the nature and precise function of their N-terminal domain linked to their helicase domain. In DnaB, this domain is defined as a collar that recruits the primase at the replication fork[33]. The N-terminal domain of RadA is formed of a C4 zinc binding motif, conserved among RadA proteins, separated from the central helicase domain by a non-conserved linker of various length (Fig. 2a). The presence of 4 cysteines in the C4 domain is required for the interaction of RadA with RecA and its oriented loading on 3-J DNA molecules. Thus, the N-terminal domains of RadA and DnaB determine their specific partnership and their distinct biological functions.

The functional and structural properties of RadA collectively support an original model of DNA branch migration driving the extension of ssDNA invasion during HR. This mechanism is based on the coordinated actions of RecA and RadA, in three successive ATP-dependent steps (Fig. 6a): Initially RecA assembles as a filament on ssDNA and catalyses strand invasion within an homologous duplex DNA (step i); via its interaction with RecA, RadA favourably gains access to both strands of the recipient dsDNA (step ii); once loaded on those two ssDNA templates, RadA translocates away from the D-loop and further unwinds the recipient dsDNA. The invading donor ssDNA is thus able to pair with the complementary strand exposed by RadA, which couples dsDNA unwinding and rewinding in a coordinated manner. By promoting DNA migration in such a way, RadA efficiently relays RecA on the 3′ side of the HR product. This stimulatory effect toward the 3′ end of the donor ssDNA on RecA-directed recombination was reported for RadA in *E. coli* in an *in vitro* assay[18]. Importantly, RadA will act similarly and symmetrically on the 5′ side of the D-loop, where RecA is unable to incorporate the complementary invading strand from the heteroduplex intermediate. RadA maximizes the incorporation of homologous internalized ssDNA molecules into the genome over long distances in both 5′ and 3′ directions.

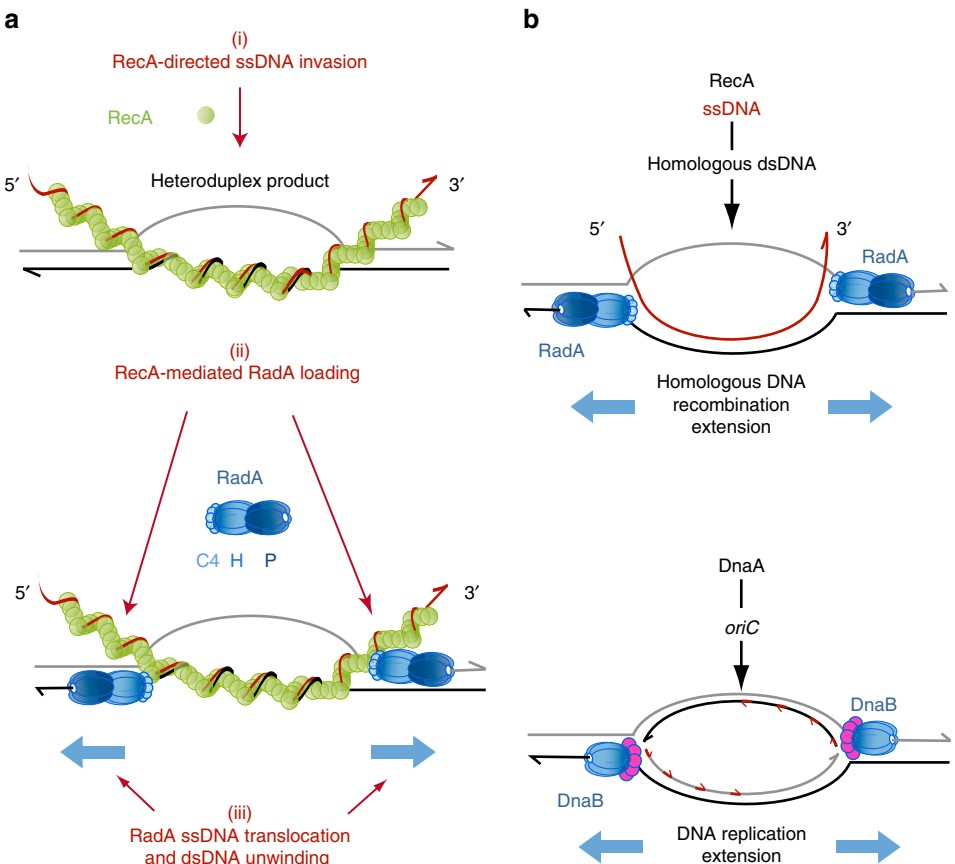

**Figure 6 | Model of HR D-loop extension promoted by RadA.** (**a**) Schematic representation of the three consecutive steps that lead to the ATP-dependent, RecA-mediated functional loading of two RadA rings on the two recipient ssDNA strands of the D-loop. (**b**) Parallel between RadA-promoted extension of a RecA-driven D-loop (upper part) and DnaB-promoted ssDNA replication at *ori*C initiated by DnaA (lower part).

Interestingly, this ATP-dependent RecA-mediated dual loading of RadA on an HR D-loop closely parallels the ATP-dependent DnaA-mediated dual loading of the replicative helicase DnaB at the replication origin of bacterial chromosomes, *ori*C[34] (Fig. 6b). Through their divergent translocation along ssDNA coupled to dsDNA unwinding, both DnaB and RadA extend the DNA bubble onto which they have been dually loaded. In both cases, their activity promotes ssDNA incorporation into the genome, via replication for DnaB and via HR for RadA.

Along with RuvB and RecG, RadA emerges as a third translocase/helicase promoting DNA branch migration in bacterial HR. RuvB and RecG differ from RadA in two major aspects. First, both translocate along dsDNA, RecG as a monomer on dsDNA and RuvB as a hexameric ring encircling dsDNA; by contrast, RadA moves along ssDNA as a hexamer. Second, they interact with different partners that target them to distinct branched DNA substrates: RuvB with RuvA, to form the RuvAB complex acting at Holliday junctions[9]; RecG with SSB at branched DNA substrates harbouring ssDNA[10]; RadA with RecA at 3-J substrates, as found in this study. An important perspective will be to elucidate how the ssDNA translocating and dsDNA unwinding activities of RadA are coordinated with the dsDNA rewinding reaction that occurs at the edges of the D-loop for extending ssDNA invasion.

In conclusion, the role of RadA that our results reveal, to expand incorporation of invading ssDNA into the recipient genome bidirectionally at HR D-loops, raises the question of how generally this role is played in the genome maintenance HR pathways to which it participates. Future work will address this

key issue, the structural and functional analysis reported here for RadA from *S. pneumoniae* providing an important mechanistic framework for conducting such an analysis.

## Methods

**S. pneumoniae transformation assays.** Stock cultures were routinely grown at 37 °C in Todd–Hewitt medium + 0.5% Yeast extract (THY) or C + Y medium to OD$_{550}$ 0.4; after addition of 15% (vol/vol) glycerol, stocks were kept frozen at −80 °C. Pre-competent wild type cultures grown at 37 °C in C + Y medium to OD$_{550}$ 0.1 were incubated with 50 ng ml$^{-1}$ synthetic competence stimulating peptide (CSP) for 7 min at 37 °C to induce competence. DNA (concentrations indicated on figures) was then added followed by 20 min incubation at 30 °C. Transformants were selected by plating on 10 ml CAT-agar supplemented with 4% (vol/vol) horse blood, followed by selection using a 10 ml overlay of CAT-agar containing selective antibiotic, after phenotypic expression for 120 min at 37 °C. Depending on the donor DNA, the selective antibiotic was rifampicin (Rif; 2 µg ml$^{-1}$), novobiocin (Nov; 4 µg ml$^{-1}$) or streptomycin (Sm; 200 µg ml$^{-1}$). Approximately 4,000 bp PCR fragments were amplified from strain R304 chromosomal DNA[35] with a mutation (M) either at the middle (PCRc) or near the end (PCRe) of the DNA fragment (Supplementary Table 1). PCR transformation experiments were performed with independent amplifications leading to variability in transformation frequencies observed. Nevertheless, the order of magnitude of the PCRc/PCRe ratios remained comparable between each experiment.

**Bacterial two-hybrid assay (BacTH).** A modified protocol of the adenylate cyclase-based bacterial two-hybrid technique was used[36]. Briefly, the isolated T18 and T25 catalytic domains of the *Bordetella* adenylate cyclase were fused to the N and C-termini of the proteins to be tested. After transformation of the two plasmids producing the complementary fusion proteins into the reporter BTH101 strain, plates were incubated at 37 °C overnight. One or more colonies for each transformation were inoculated into 3 ml of LB medium supplemented with ampicillin and kanamycin to O.D$_{600}$ 0.6. 2 µl of each culture were dropped onto LB plates supplemented with ampicillin, kanamycin, streptomycin, 1 mM IPTG and

10% X-Gal and incubated for 24 h at 25 °C. Each transformation was done at least in triplicate and a representative result is shown.

**Recombinant plasmids.** Recombinant plasmids expressing pneumococcal RadA and RecA proteins in *E. coli* were obtained after cloning of *radA* and *recA* sequences from the genome of pneumococcal R800 derivative strain into pET21d and pKHS (a pET28 derivative)[37] respectively to obtain pET-*radA* and pKHS-*recA* vectors. *radA*$_{AC4}$ and *radA*$_P$ were cloned by FastCloning[38] into the IBA13 plasmid in frame with a thrombin cleavable StrepII tag.

**In vitro site-directed mutagenesis.** All the point mutations in *radA* (*radA*$_{C27A}$, $_{K101A, K251A, R253A}$) and *recA* (*recA*$_{K85A}$) were introduced by site-directed mutagenesis PCR on the recombinant plasmids, pET-*radA* and pKHS-*recA* accordingly. We used an adapted protocol from the Stratagene 'QuickChange' with only one mutator primer carrying the modified sequence for each mutation (Supplementary Table 1), and with both the *Pfu turbo* polymerase (Stratagene) and the *TAQ ligase* (New England Biolabs) in the PCR reaction.

**radA mutagenesis in S. pneumoniae.** Point mutations of RadA were introduced at the native *radA* locus in R1818 by using pET-*radA*$_{K101A}$, pET-*radA*$_{K251A}$ and pET-*radA*$_{R253A}$ plasmids as transforming donor DNA. Pre-competent cultures (1.9–3.7 × 10$^8$ cfu ml$^{-1}$) grown at 37 °C in C + Y medium to OD$_{550}$ 0.1 were incubated with 50 ng ml$^{-1}$ synthetic competence stimulating peptide (CSP) for 7 min at 37 °C to induce competence. About 10 ng ml$^{-1}$ mutant plasmids were added, followed by 20 min incubation at 30 °C. Fourteen volumes of C + Y medium were added followed by 2 h incubation at 37 °C. Approximately 100 μl cultures were plated on CAT-agar supplemented with 4% (vol/vol) horse blood. For each mutant, clones were screened by PCR on liquid cultures.

The *radA*$_{C27A}$ mutant was constructed using SOEing PCRs: PCR1 and PCR2 were directly amplified from pneumococcal strain and were used as templates in the final PCR (2 kb) using the outer Olea23 and Olea19 primers (Supplementary Table 1) before transformation of pre-competent R1818 cells.

**Immunodetection of RadA by western blot.** Uncropped scans of important gels are shown in Supplementary Fig. 9. Approximately 60 μl pre-competent cultures grown at 37 °C in C + Y medium to OD$_{550}$ 0.1 were incubated with 25 ng ml$^{-1}$ synthetic competence stimulating peptide (CSP) for 7 min at 37 °C to induce competence. Cultures were centrifuged 5 min at 5,000 g 4 °C, supernatants were discarded and pellets were stored 12 h at −80 °C. Pellets were resuspended with 50 μl buffer (TE 1X and 0.01 Doc) and heated 5 min at 80 °C with loading buffer. 10 μl extracts were then loaded on SDS-PAGE precast 4–20% gels (Mini-PROTEAN TGX gel, Bio-Rad) followed by electrophoresis at 180 V for 35 min in 1X TGS buffer. After migration, proteins were transferred to a nitrocellulose membrane (Mini Nitrocellulose Transfer Packs, Bio-Rad) with the trans-Blot Turbo Transfer System (Bio-Rad). The nitrocellulose membrane was then cut above the 17 kDa marker band. The top part of the membrane was probed with anti-RadA antibodies (1:1,000) and the bottom part with anti-SsbB (1:5,000). Both primary antibodies were raised in rabbit (Eurogentec) from purified protein.

**Protein expression and purification.** All the *S. pneumoniae* proteins were expressed in *E. coli* BL21-Rosetta(DE3)-pLysS (Cm$^R$) cells (Novagen) and were purified following different procedures according to the type of experiment. In both strategies, proteins were over-expressed in *E. coli* and purified as soluble proteins.

**Purification procedures for analysis of RadA and RecA.** RadA, RecA and all derivatives were purified as native proteins, without any tag. Expression clones containing recombinant pET-*radA* (Amp$^R$), pKHS-*recA* (Kan$^R$) and all derivatives were grown at 37 °C in 1 l LB/Cm (10 μg ml$^{-1}$)/Amp (100 μg ml$^{-1}$) or Kan (50 μg ml$^{-1}$) medium until the OD$_{600}$ reached 0.7. Recombinant protein expression was then induced by adding 0.5 mM IPTG followed by growth at 37 °C for 4 h for RadA proteins and 2 h for RecA proteins. Cells were collected by centrifugation, resuspended in sucrose buffer (25 mM Tris–Cl $_{pH 8}$, 25% sucrose, 1 mM EDTA, 200 μg ml$^{-1}$ lysozyme) before liquid nitrogen freezing. Cell pellets were next thawed overnight on ice and suspensions were clarified by ultra-centrifugation (Ti50.2 rotor, at 146,542 g 4 °C for 1h30). For RadA proteins, the clarified supernatant was further fractionated by two successive transient precipitations with ammonium sulfate (55% and then 35%). Pellet was re-solubilised and dialysed in 50 mM Tris–Cl $_{pH 8}$, 100 mM NaCl, 5% glycerol before further purification by FPLC (Äkta purifier-10, GE Healthcare) with a Hi-Trap Q HP 1 ml column followed by a Hi-Trap heparin HP 1 ml column (GE Healthcare). Both columns were equilibrated in 100 mM NaCl, 50 mM Tris–Cl $_{pH 8}$, 5% glycerol, washed after proteins loading with 150–200 mM NaCl, and RadA eluted with a linear gradient of 0.2 to 1 M NaCl. The final fractions of RadA proteins were stored at −20 °C in 300 mM NaCl, 25 mM Tris–Cl $_{pH 8}$, 45% glycerol, 1 mM TCEP (Tris(2-carboxyethyl)phosphine)-HCl $_{pH 7}$. Further analysis of RadA oligomerization was performed by gel filtration. Purified RadA wild type or mutant proteins were sequentially injected and separated on a Superdex 200 increase 10/300

analytical sizing column (GE healthcare) equilibrated with 25 mM Tris $_{pH 8}$, 200 mM NaCl, 20% glycerol. The column was calibrated with 43, 158 and 440 kDa standard proteins (Ovalbumine, Aldolase and Ferritine respectively) in the same buffer (Supplementary Fig. 4b). For RecA proteins, the clarified supernatant was transiently precipitated by addition of 0.1% Polymin P. After several washes at 200 mM NaCl, 20 mM Tris–Cl $_{pH 7.6}$, 0.5 mM TCEP, the protein fractions containing RecA were resolubilized in 20 mM Tris–Cl $_{pH 7.6}$, 1 M NaCl, 0.5 mM TCEP. The protein solution was then mixed with 0.23 g ml$^{-1}$ ammonium sulfate and stirred for 30 min at 4 °C. After centrifugation in a JA.20 rotor, at 12,000 g for 10 min at 4 °C, the supernatant containing RecA was consecutively loaded on Phenyl, Heparin and Q columns (HiTrap columns, GE Healthcare) and purified by FPLC. RecA proteins were stored at −20 °C in 300 mM NaCl, 10 mM Tris–Cl $_{pH 7.6}$, 45% glycerol, 1 mM TCEP. The concentrations of the purified wild type and mutant proteins were determined by ultraviolet absorbance, Bradford assay, SDS–polyacrylamide gel electrophoresis (SDS–PAGE) and coomassie staining analysis (Supplementary Fig. 4a).

**RadA purification for structural analysis.** The pET-*radA* and IBA13-*radA* plasmids were transformed in *E. coli* BL21-Rosetta(DE3), and grown at 37 °C in Terrific Broth (TB). At an O.D$_{600}$ of 0.8, protein expression was induced either with 1 mM IPTG or 2 μg ml$^{-1}$ AHT (IBA) respectively and cells were grown overnight at 16 °C. Cells were harvested by centrifugation and frozen in liquid nitrogen. Cells were resuspended in Lysis buffer containing 50 mM HEPES $_{pH 8}$, 1 mM DNase, 2.5 mM MgCl$_2$ and protease inhibitors (Roche), and lysed by Emulsiflex at 15,000 psi three times. The StrepII-RadA proteins were purified by using a StrepTrap column (GE healthcare). Proteins were eluted from the column with 2.5 mM desthiobiotin and subsequently loaded on and eluted from a Superdex 200 (GE healthcare), equilibrated and run with 50 mM Tris $_{pH 8}$, 150 mM NaCl and 5 mM MgCl$_2$. Selenium labelled proteins were transformed in T7 Express Crystal cells (New England Biolabs) and grown in SelenoMet Medium (Molecular Dimensions) and induced at O.D$_{600}$ 0.6.

**Analysis of RadA molecular mass.** To determine the molecular mass of RadA$_{FL}$ and RadA$_P$, multi-angle light scattering (MALS, DAWN HELEOS II, Wyatt Technology) measurements combined with refractive index detection (Optilab t-rEX, Wyatt Technology) were performed[39]. The samples at a concentration of 2 mg ml$^{-1}$ were injected onto a Superdex200 Increase (GE healthcare) coupled to a MALS detector. The flow rate was set to 0.4 ml min$^{-1}$ with an injection volume of 50 μl and the refractive index signal was collected at 298 K.

**X-ray fluorescence analysis.** The X-ray fluorescence (XRF) analysis was performed on beamline ID30-B at the European Synchrotron Radiation Facility using a Rontec Silicon drift diode XRF detector.

**RadA crystallization.** RadA proteins were concentrated in Amicon Ultra centrifugal filters to above 3 mg ml$^{-1}$. Crystallization trials were performed using the sitting drop vapour diffusion method at 18 °C using a 200 nl dispensing robot (Mosquito). Several conditions were obtained that induced the crystallization of the full-length and truncated proteins. RadA$_P$ crystallized best in 9–12% PEG4000 and 100 mM Tris with 30% EG used as a cryoprotectant for flash freezing at 100 K. Crystals appeared within a few hours but were left on the plate for at least a week before harvesting. RadA$_{FL}$ crystallized in 200 mM magnesium formate, 100 mM HEPES $_{pH 8}$ and 5% glycerol with the final concentration of glycerol being 20% for flash freezing of the crystals.

Data sets for the native and SeMet were collected at 0.9796 Å on Proxima-1 beamline at the SOLEIL synchrotron facility equipped with a Pilatus detector. RadA$_P$ crystals belong to the R3 space group and contain two monomers per asymmetric unit. Data processing was done with XDS[40] and XSCALE[41], and scalepack in the ccp4 suite of programs was used to export the scaled data[42]. HySS was used to find the sites by single anomalous dispersion (SAD) of the Selenium atoms using 3.5 Å as the highest resolution and autosol was used to obtain an initial model automatically, both from the Phenix suite of programs[43]. The dataset of a native crystal was solved by molecular replacement (MR) using Phaser[44] to a resolution of 2.5 Å. Refinement of the selenium and the native datasets were performed using phenix.refine followed by rounds of manual model building with Coot[45]. The RadA$_{FL}$ crystals diffracted at best to 3.5 Å and belong to the I222 space group. Five datasets were collected from one crystal of Selenium-containing RadA$_{FL}$ and merged together using XDS[41] and XSCALE[41]. The protease domain structure was used for MR in Phaser[44] from the ccp4 suite[42] and gave a partial solution. The RecA ATPase domain was used as the second ensemble in the MR search after the placement of the protease domain. The model was built manually using Coot and an alignment file of RecA and RadA was used to assign amino acids. To improve the Rfree, the SAD Selenium sites were located using ShelxD[46] and input in Phaser[44] to refine the preliminary model. Several rounds of refinement with NCS constraints with Buster[47], phenix_rosetta.refine[48] and refmac5 (ref. 49) (with jelly body and ProSMART activated) followed by manual building by Coot were required. Validation of the structures was performed by using Molprobity[50]. The figures of the structures were obtained with Chimera[51]

and Coot[45]. The RadA$_{\Delta C4}$ protein crystallized with a space group I121, in 16% PEG4000, 200 mM Na Acetate, 100 mM Tris $_{pH\ 8.5}$ and was frozen in the reservoir buffer supplemented with 30% EG. The data were processed by XDS[40] and XSCALE[41] and the crystal packing was obtained using the full-length trimer as a model for molecular replacement. A quick refinement was performed with refmac5 (ref. 49) to confirm the molecular replacement solution was correct.

**Electron microscopy.** StrepII tagged RadA ($\sim 0.04$ mg ml$^{-1}$) in 50 mM KCl, 0.5 mM DTT, 10 mM Tris–HCl $_{pH\ 7.5}$, 2 mM MgAc, 2 mM ATPγS was incubated at 37 °C for 30 min with 75 ng ssDNA from the M13mp18 lac phage (NEB). Four microliters of DNA-bound or free RadA ($\sim 0.01$ mg ml$^{-1}$) were adsorbed to glow-discharged carbon-coated copper grids (EMS) for 1 min, blotted, and stained with 2% uranyl acetate. The grids were imaged on a Tecnai T12 (FEI) equipped with an Ultrascan camera.

**ATPase assay.** A coupled spectrophotometric enzyme assay was used to measure the ATPase activities of RadA wild type and mutants as follows[52,53]. The regeneration of ATP from phosphoenolpyruvate and ADP, coupled to the oxidation of NADH, was observed as a decrease in absorbance at 340 nm. 600 nM of wild type or mutants RadA proteins, ATP (ranging from 0 to 2 mM) and 2 μM nucleotides (ssDNA phiX174; Biolabs) or 4 μM nucleotides (dsDNA phiX174-RF1; Biolabs) were used to initiate the reactions.

The ATPase assays were carried out on 96-well plates at 37 °C with a Thermo Scientific Varioskan Flash spectral scanning multimode reader. The reactions were carried out in a buffer containing 10 mM Tris–Cl $_{pH\ 7.5}$, 4 mM Mg(OAc)$_2$, 0.1 mM DTT, 1 mM phosphoenolpyruvate, 0.1 mg ml$^{-1}$ NADH, 100 unit ml$^{-1}$ pyruvate kinase, 10 unit ml$^{-1}$ L-lactic dehydrogenase. The NADH extinction coefficient at 340 nm of 6.22 mM$^{-1}$ cm$^{-1}$ was used to convert the amount of NADH oxidized to the amount of ATP hydrolysed. The graph and table were constructed using a Michaelis–Menten model on three experiments with Prism-GraphPad 7 software. Best-fit and standard error values are presented on Supplementary Fig. 4c.

**DNA substrates for helicase assays.** DNA substrates were generated by using synthetic oligonucleotides and/or long circular ssDNA M13mp18 (Biolabs) (Supplementary Table 1 and Supplementary Fig. 8). Equimolar concentrations of 2 or 3 complementary ssDNA molecules were mixed in annealing buffer (20 mM Tris–Cl $_{pH\ 7.6}$, 100 mM NaCl, 10 mM Mg(OAc)$_2$, 1 mM DTT), heated for 5 min at 100 °C, placed in boiling water and allowed to cool to room temperature. The schematic representation of the structures and their migration profiles on 4% acrylamide (29:1)/0.3 X TBE gel are shown in Supplementary Fig. 6d. One strand of the various DNA structures was labelled with a Cy3 fluorochrome at the 5′ or at the 3′ end in order to be further detected by a fluor imager (Typhoon Trio, Fuji-GE Heathlcare) with an Abs/Em at 532/580 nm.

**Helicase assay.** Increasing amounts of RadA (80–750 nM) were incubated 15 min at 37 °C with 10 nM of various short Cy3 labelled DNA substrates (Supplementary Fig. 8) in 10 μl solutions containing 10 mM Tris–Cl $_{pH\ 7.5}$, 0.1 mg ml$^{-1}$ BSA, 8% glycerol, 1 mM DTT, 50 mM NaCl, 10 mM Mg(OAc)2 and 5 mM ATP. In order to quench the reaction and to deproteinize the DNA, 0.1% SDS and 1 mM EDTA were added to the reaction for 5 min at 37 °C before loading on 4% acrylamide (29:1)/0.3× TBE gels. Cy3 labelled oligonucleotides were detected with the Typhoon Trio imager.

In the presence of both RecA and RadA proteins, 150–900 nM of RecA (wild type or mutant) and 750 nM of RadA (wild type or mutants) were added simultaneously to the reactions.

For the helicase assays performed with long circular ssM13mp18 substrates (Supplementary Fig. 8), 0.5–2 μM of wild-type and mutant RadA were incubated 15 min at 37 °C with about 6 nM DNA substrate in 5 μl reactions. After deproteinization, 0.5 μl of buffer (xylene cyanol, glycerol 50%) were added to the reactions before loading on 1.2% agarose/TAE 1× gel. The fluorescence detection was managed, as described above.

**Quantification of helicase products.** The DNA products in % of total DNA have been quantified by using the Multigauge software. The experiment performed without any protein has been retracted to each experiment. The % of helicase products relative to the initial structures reported below the gels. Helicase assays were performed at least twice independently and led to reproducible results. In particular, experiments performed with wild-type proteins on the two 3-J substrates were reproduced between three and five times and resulted in the same ratio of products at the protein concentrations tested. Experiments performed with RecA and RadA point mutants were done at least twice independently. One typical experiment is presented in each case. Importantly, we used at least two distinct purified native protein preparations to certify the reproducibility of the experiments.

**DNA binding assay on ssDNA and dsDNA.** Increasing amounts of RadA (50–750 nM) were incubated 15 min at 37 °C with 10 nM of Cy3 labelled DNA (100-mer) and either ssDNA or dsDNA (S or D structures in Supplementary Fig. 8)

in a 10 μl reaction solution containing 50 mM Tris–Cl pH 7.5, 0.1 mg ml$^{-1}$ BSA, 8% glycerol, 1 mM DTT, 100 mM NaCl, 10 mM Mg(OAc)$_2$, with or without 5 mM ATP (GE Healthcare). Samples were loaded on a 4% acrylamide (29:1)/0.3 × TBE gel and submitted to electrophoresis ≈ 10 V cm$^{-1}$ for 1 h in 0.3 × TBE buffer. Free DNA and nucleoprotein complexes were then directly detected on the gel using the Typhoon Trio (Supplementary Fig. 5). In parallel, the binding profiles of RadA, RadA$_{K251A}$ and RadA$_{R253A}$ (150–600 nM) were also tested with 10 nM of γ33 P- labelled oligonucleotide (100-mer) in the same reaction conditions, as previously described. After electrophoresis, the gel was dried and radioactivity detected with a phosphorimager.

The competition experiment in Supplementary Fig. 5b was conducted with 750 nM RadA incubated 15 min at 37 °C with 10 nM Ovio10-Cy3 in a reaction containing 10 mM Tris–Cl $_{pH\ 7.5}$, 0.1 mg ml$^{-1}$ BSA, 8% glycerol, about 50 mM NaCl and 10 mM Mg(OAc)$_2$. At the same time as Ovio10-Cy3 or 10 min later, 5, 10 or 20 nM of Ovio8-Cy5 were added to the reaction and incubated 10 min. The solutions were then loaded on a 4% acrylamide (29:1)/0.3 × TBE gel and submitted to electrophoresis as previously described. The fluorescent DNAs were then visualized with Typhoon Trio with Abs/Em of 532/580 nm (Cy3) and 633/670 nm (Cy5).

**Data availability.** All data within the article and its Supplementary Information files are available from the authors.

Coordinates and structure factors have been deposited in the Protein Data Bank under accession numbers 5LKM and 5LKQ for RadA$_{FL}$ and RadA$_P$ respectively.

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

## Acknowledgements

We thank Calum Johnston, Nathalie Campo, Dave Lane and Jean-Yves Bouet for critical reading of the manuscript. This work was funded by the Centre National de la Recherche Scientifique, University Paul Sabatier, Agence Nationale de la Recherche (grant ANR-10-BLAN-1331) and the European Research Council (MolStructTransfo, 281149). L.M. was supported by a PhD grant from Ministère de la Recherche et de l'Enseignement. We thank Pierre Legrand and tutors at the 2014 Diamond-CCP4 course for advice on data collection, processing and refinement; the staff at Soleil, the ESRF and DLS for beamtime and the platform of Crystallography at the Institut Pasteur for technical support. We thank Annick Dujeancourt and Francesca Gubellini for help with cloning of BacTH constructs.

## Authors contributions

R.F., P.P. and C.R. designed the experiments. L.M. and M.B. constructed the pneumococcal strains and performed the transformation experiments. L.M., V.M. and C.R. assisted by C.G. and A.-L.S. purified recombinant RadA derivatives. L.M. and V.M. performed RadA ATPase and helicase assays, and EMSA analysis. C.R. performed all the structural analyses assisted for the crystallography by H.R. C.R. assisted by T.P. performed the Bacterial Two-hybrid study. C.R. performed the molecular mass determination by MALS. P.P., R.F, C.R. and L.M. wrote the manuscript with contributions of V.M. and M.B.

## Additional information

**Competing interests:** The authors declare no competing financial interests.

