## [Peer Review File · Nature Communications]

Reviewers' Comments:

Reviewer #1 (Remarks to the Author)

The present manuscript by Marie et al. provides a structural and biochemical analysis of the bacterial RadA protein and its role in promoting homologous recombination (HR). RadA is shown to consist of an intriguing fusion between a RecA-family ATPase domain and a catalytically-defunct Lon-type protease domain, a fold normally found in conjunction with AAA+ ATPases. Solution and crystallographic studies provide compelling evidence that RadA assembles into a hexamer most closely akin to SF4-family helicases, rather than forming filamentous oligomers as do the related RecA HR recombinases. This similarity is demonstrated to extend to an ability to unwind both DNA duplexes and D-loop branch point mimics with a 5'-3' polarity. Interestingly, RecA is shown to interact with RadA and to differentially regulate the unwinding activity of the helicase on the two classes of DNA junctions that would be found in a D-loop structure.

Overall, this was an enjoyable and convincing read. The paper is crisply written, the experiments are straightforward and well-controlled, and the conclusions are plausibly derived from the reported data. Although a recent published study has already noted the link between RadA and RecA, the current work goes further, showing that RadA is a DnaB-like helicase and thereby overturning the idea that RadA might operate like RecA in promoting branch migration. The findings additionally suggest that the RecA interaction with RadA may help to promote "backward" (upstream) D-loop expansion while somewhat repressing the natural tendency of a 5'-3' unwinding activity to promote forward progression, possibly as a means to promote expansion of the invaded strand more equally in a bidirectional manner. These findings should resonate with a reasonably broad audience ranging from helicase aficionados to replication/repair types. Pending resolution of a few minor issues, publication in Nature Communications would see well warranted.

Specific comments:

--Fig. 6. The DNA unwinding assays in Fig. 5 are not necessarily the best ones to probe the suggested model. The ssDNA in a D-loop, which is wrapped around the paired strands will have a much different level of accessibility than an ssDNA tail. Probing the effect of RadA on a D-loop formation/extension assay would be useful to further test the proposed model.

--It seems that the C2 form of RadFL is similar to the C2 form of T7 gp4 (Wigley and coworkers). It would be useful to acknowledge this point and to include a supplemental structural analysis of the similarities and differences between two hexamers.

--Fig. 3A legend. What is the rmsd similarity between T7gp4 and the RadA ATPase?

--Fig. 4C. How stable is the RadA hexamer? How do the authors envision that it can load around a circular DNA?

--Fig. ED2. If class averages are going to be used to make claims about particle symmetry, then multiple classes, approximate particle frequency, and eigen images should be shown as well.

--Fig. ED2C legend. Please state the expected molecular weights of each hexamer in comparison to the measured values.

--Fig. ED3E. The arginine finger modeling shown here doesn't add much to the paper and should be removed.

--Fig. ED4A. Presumably the C27A mutant forms a non-physiological disulfide-bonded dimer between cysteines in the disrupted Zn-finger, rather than a physiological dimer (which could be

tested by including a strong reducing agent (e.g., 10 mM DTT or 1 mM TCEP in the gel and loading buffer). This should be clarified.

Reviewer #2 (Remarks to the Author)

This paper presents a lot of data on the activity of RadA/Sma, including a crystal structure, that will be useful in understanding its mechanism. However, the results do not definitively support the model as shown in Figure 6.

A general deficiency of the paper is that statistics are not stated. The number of repeats are not known and for quantified data, there is no error reporting. Some of the differences are quite subtle (e.g. Figure 5) and one really needs to know the confidence in the values. Moreover, some quantitation is not shown (Figure 5h, i) that is needed for comparison.

The key result of the paper would be the knowledge of the interaction between RadA and RecA and that this interaction focuses the helicase activity of RadA to movement on fork substrates (essentially figure 6). However, the data at this stage is a little preliminary and phenomenological, and does not confidently exclude other possibilities.

Firstly, the interaction between RecA and RadA is not explored further. (An)Other measurement(s) of the interaction are needed, such as pull-down/SEC/SEC-MALS/native-MS.

Secondly, the data in Figure 5 are not wholly convincing and do not fully explore the possible mechanism. One control that is needed is to repeat the helicase assays of Figure 4 with the corresponding 30 nt overhang, to show that this is the reason for the inefficiency of unwinding of the 3-J1 substrate. The general effect of RecA is to inhibit the 5-3 unwinding activity of RadA that loads onto the 5'-terminated overhang. The increase in the F1a or F2a products (the RecA activation) is rather modest (particularly if errors were to be shown) compared to the inactivation of the unwinding that produces the S1 or F2b products. The results in h/I are not quantified and would be better repeated on 3-J2 which produces more measurable product.

It is not clear on the 3-J1 and 3-J2 substrates why the loading would be on the "bottom" strand and go in the "5' direction" (these terms are confusing –see below). Why wouldn't some loading events be directed to the "top strand" and go in the "5' direction"? The 3-J substrates could be said to mimic both junctions (as shown in Figure 5b). It could be that this IS happening but this result is hidden by the background of helicase activity. The authors need to design different substrates/more sophisticated assays to fully explore the model.

The authors make a lot of effort to make the paper understandable. However, it is still confusing to follow and in particular Figure 5 switches between discussion of 5-3 helicase activity and 5' direction unwinding, which is still 5-3 helicase activity.

Itemized Responses to Reviewer's comments

Our point-by-point responses to the comments/questions of the Reviewers are inserted below, in blue, in the body of their reviews, next to each comment.

Reviewers' comments:

Reviewer #1 (Remarks to the Author):

The present manuscript by Marie et al. provides a structural and biochemical analysis of the bacterial RadA protein and its role in promoting homologous recombination (HR). RadA is shown to consist of an intriguing fusion between a RecA-family ATPase domain and a catalytically-defunct Lon-type protease domain, a fold normally found in conjunction with AAA+ ATPases. Solution and crystallographic studies provide compelling evidence that RadA assembles into a hexamer most closely akin to SF4-family helicases, rather than forming filamentous oligomers as do the related RecA HR recombinases. This similarity is demonstrated to extend to an ability to unwind both DNA duplexes and D-loop branch point mimics with a 5'-3' polarity. Interestingly, RecA is shown to interact with RadA and to differentially regulate the unwinding activity of the helicase on the two classes of DNA junctions that would be found in a D-loop structure.

Overall, this was an enjoyable and convincing read. The paper is crisply written, the experiments are straightforward and well-controlled, and the conclusions are plausibly derived from the reported data. Although a recent published study has already noted the link between RadA and RecA, the current work goes further, showing that RadA is a DnaB-like helicase and thereby overturning the idea that RadA might operate like RecA in promoting branch migration. The findings additionally suggest that the RecA interaction with RadA may help to promote "backward" (upstream) D-loop expansion while somewhat repressing the natural tendency of a 5'-3' unwinding activity to promote forward progression, possibly as a means to promote expansion of the invaded strand more equally in a bidirectional manner. These findings should resonate with a reasonably broad audience ranging from helicase aficionados to replication/repair types. Pending resolution of a few minor issues, publication in Nature Communications would see well warranted.

We are delighted to read that Reviewer 1 shares our enthusiasm about our genetic, structural and biochemical characterization of pneumococcal RadA and finds our study well warranted for publication in *Nature Communications*.

Institut Européen de Chimie et Biologie

2, rue Robert Escarpit - 33607 Pessac, France
Tél. : +33(0)5 40 00 30 38 - Fax. : +33(0)5 40 00 22 15
www.iecb.u-bordeaux.fr

Specific comments:

--Fig. 6. The DNA unwinding assays in Fig. 5 are not necessarily the best ones to probe the suggested model. The ssDNA in a D-loop, which is wrapped around the paired strands will have a much different level of accessibility than an ssDNA tail. Probing the effect of RadA on a D-loop formation/extension assay would be useful to further test the proposed model.

The helicase assays we used were done to probe the interplay between RadA and RecA and, indeed, they provided strong evidence that RecA mediates the loading of RadA on 3-way junctions, especially on the bottom and fully paired strand of the branched DNA molecule. This result supports the genetic evidence presented in Figure 1 and in Supplemental Figure 1, indicating that RadA acts to translocate ssDNA in both direction from the D-loop intermediate.

We agree that the branched DNA substrates used in our helicase assays are not the best substrates to probe the proposed model. In fact, we did perform D-loop assays to investigate the functional interplay between RecA and RadA. Such assays are based on the use of a supercoiled plasmid and a short homologous ssDNA molecule as recipient and donor substrates respectively. RecA alone is proficient in generating the D-loop product visualized by native gel electrophoresis following deproteinization. In the presence of RadA, we did not observe stimulation of RecA-dependent D-loop formation. In addition, in the D-loop assay we used, we cannot measure an extension of ssDNA incorporation by RadA, because the pairing is limited by the topology of the supercoiled D-loop product. Since these results are not informative, we have not included them in the manuscript. We are currently undertaking more appropriate and sophisticated assays to probe the RadA-promoted ssDNA incorporation from RecA-directed D-loop intermediates. In our opinion, this represents a whole study in its own right and therefore is beyond the scope of this structural and functional study of RadA.

--It seems that the C2 form of RadFL is similar to the C2 form of T7 gp4 (Wigley and coworkers). It would be useful to acknowledge this point and to include a supplemental structural analysis of the similarities and differences between two hexamers.

We added a panel in Figure 3a that compares the structure of the C2 form of T7 GP4 and RadA.

--Fig. 3A legend. What is the rmsd similarity between T7gp4 and the RadA ATPase?

We added this information in the legend of Figure 3a

--Fig. 4C. How stable is the RadA hexamer? How do the authors envision that it can load around a circular DNA?

RadA forms very stable hexamers. As shown in supplementary Fig2a-c and supplementary Fig4b, they can be observed even at low protein concentration by gel filtration and electron microscopy, in the absence of DNA. At these concentrations the protein is found to interact with ssDNA in a pearl necklace pattern (Figure 4f), suggesting that the hexamers encircle the ssDNA and supporting the notion that a ring-opening mechanism underlies this loading process. Crystallographic data indicate that the stability of the hexamer is provided by the electrostatic interactions between P domains. We think that in contact with DNA, RadA hexameric rings could open, allowing the loading of the RadA hexamer around the ssDNA substrate. It is possible that DNA binding motifs exist outside of the RadA ring. This external DNA binding surface could modify the electrostatic properties of one interface between two P domains and allow transient ring opening. Components of the HR machinery and, in particular, RecA could also facilitate this ring opening and loading of RadA on ssDNA. Understanding the loading mechanism of RadA on ssDNA is one of the many important perspectives of this work. However, we feel that discussing how RadA loading could proceed on ssDNA is too speculative to warrant inclusion in this manuscript.

The hexameric ring has a weak point at the interface of the B chain. The ΔG calculated by PISA is much higher at this point, indicating that it might open when necessary upon the presence of the DNA.

--Fig. ED2. If class averages are going to be used to make claims about particle symmetry, then multiple classes, approximate particle frequency, and eigen images should be shown as well.

We only used these class averages to show that both Rad_{FL} and Rad_P form hexameric particles in solution as it was observed by MALS and in the crystal structures. We removed the sentences about the "C2 symmetry" and "C6 symmetry" of these particles in the legend of supplementary figure 2 (a and b panels) since this information is not used in our conclusions.

--Fig. ED2C legend. Please state the expected molecular weights of each hexamer in comparison to the measured values.

We added this information in the legend of the supplementary figure 2. The values obtained by MALS are very close to the calculated ones.

--Fig. ED3E. The arginine finger modeling shown here doesn't add much to the paper and should be removed.

We have removed the modelling figure as suggested.

--Fig. ED4A. Presumably the C27A mutant forms a non-physiological disulfide-bonded dimer between cysteines in the disrupted Zn-finger, rather than a physiological dimer (which could be tested by including a strong reducing agent (e.g., 10 mM DTT or 1 mM TCEP in the gel and loading buffer). This should be clarified.

We agree that the extra band seen on the denaturing gel showing the purified recombinant C27A RadA mutant would most probably represent a disulfide-bonded dimer between cysteines in the disrupted Zn-finger. We show by Western blot analysis using anti-RadA antibodies that this band is a dimer of RadAC27A. We have done the experiment proposed by reviewer 1. By using high concentration of DTT and/or TCEP in the gel, the pattern seen on the denaturing gel was the same. The RadAC27A mutant displays ATPase and helicase activity. This self-association of a minor fraction of purified RadA changes the interpretation we made of the results obtained with this mutant protein, and we have adapted our interpretation accordingly.

Reviewer #2 (Remarks to the Author):

This paper presents a lot of data on the activity of RadA/Sma, including a crystal structure, that will be useful in understanding its mechanism. However, the results do not definitively support the model as shown in Figure 6.

First, we would like to mention that the model in Figure 6 is a working model integrating the genetic, structural and biochemical results obtained to understand the molecular mechanism of RadA. Furthermore, as explained below, we added new biochemical experiments that further support this model, on the basis of the concerns raised.

A general deficiency of the paper is that statistics are not stated. The number of repeats are not known and for quantified data, there is no error reporting. Some of the differences are quite subtle (e.g. Figure 5) and one really needs to know the confidence in the values. Moreover, some quantitation is not shown (Figure 5h, i) that is needed for comparison.

All the quantifications have been added below each gel. We have clearly mentioned in a dedicated paragraph in the Methods section of the revised manuscript how the helicase assays have been quantified. The helicase assays have been reproduced several times independently, with at least two distinct protein preparations. However, the number of repeats was identical for all experiments and mentioning this numbers in the figure legends would overload them. Instead, this has been clearly mentioned as follows in the Methods section:

Helicase assays were performed at least twice independently and led to reproducible results. In particular, experiments performed with wild-type proteins on the two 3-J substrates were reproduced between 3 and 5 times and resulted in the same ratio of products at the protein concentrations tested. Experiments performed with RecA and RadA point mutants were done at least twice independently. One typical experiment is presented in each case. Importantly, experiments were reproduced by using at least two distinct purified native protein preparations, to certify their reproducibility.

The key point of the helicase assays is the unwinding of the 3-J substrates in the 5' direction, an activity that requires both wild-type RecA and RadA. This main result is clearly observed several times in the gels presented in Figure 5 and in supplementary Figure 6. The experiments performed with the point mutants validate the conclusion that 3-J unwinding in the 5' direction relies on the specific interplay between RadA and RecA on DNA.

The key result of the paper would be the knowledge of the interaction between RadA and RecA and that this interaction focuses the helicase activity of RadA to movement on fork substrates (essentially figure 6). However, the data at this stage is a little preliminary and phenomenological, and does not confidently exclude other possibilities.

We agree that the interaction between RadA and RecA is a key result of the paper but it is not the sole one. As mentioned by referee # 1, our study also shows that “...RadA is a DnaB-like helicase and thereby overturning the idea that RadA might operate like RecA in promoting branch migration”. The physical and functional interaction between RecA and RadA furthers our understanding of the role of RadA in HR, which we show extends incorporation of invading ssDNA into the recipient genome.

Firstly, the interaction between RecA and RadA is not explored further. (An)Other measurement(s) of the interaction are needed, such as pull-down/SEC/SEC-MALS/native-MS.

We attempted to isolate a RecA-RadA complex either after co-expression and pull-down of the two proteins in *E. coli*, or by mixing both purified proteins *in vitro* to assay their association by pull-down or by co-immunoprecipitation (Co-IP). So far, none of these attempts have successfully isolated a RadA-RecA complex. Importantly, the Co-IP experiment was validated by using DprA, a known partner of pneumococcal RecA. We did not mention these experiments in the manuscript, since they were negative.

They indicate that the physical interaction between these two proteins is very dynamic, which complicates its further biochemical characterization. The full characterization of the physical interaction of RadA with RecA is a major perspective of this work, and in our opinion, is also beyond the scope of this study.

Secondly, the data in Figure 5 are not wholly convincing and do not fully explore the possible mechanism. One control that is needed is to repeat the helicase assays of Figure 4 with the corresponding 30 nt overhang, to show that this is the reason for the inefficiency of unwinding of the 3-J1 substrate.

Indeed, in the helicase assays performed on the 3J-1 and 3J-2 substrates, it was not possible to conclude whether the difference in S1 and F2-b products respectively is due to the difference in the length of the ssDNA tail onto which RadA loads or in the length of dsDNA to unwind (as we mentioned in the text). To clarify this point, we compared RadA efficiency to unwind a duplex of 30 bps or of 70 bps in a M13-based helicase assay (as in Figure 4c). The results presented in supplementary Figure 4d of the revised version of the manuscript clearly show that RadA was equally efficient in unwinding both duplexes.

Having excluded the second hypothesis that the length of the dsDNA to unwind explains the difference in S1 and F2-b products, we can infer that it is the length of ssDNA onto which RadA loads that determines the difference in the yield of unwound S1 and F2-b products in helicase assays performed on 3J-1 and 3J-2, respectively.

This experiment is presented as follow in the results section of the revised version of the manuscript, in the section entitled “**Interplay between RecA and RadA at three-way DNA junctions**”:

Helicase assays comparing unwinding efficiency of RadA targeting a 30 bp and a 70 bp duplex associated to a large circular ssDNA molecule strongly support the latter possibility (Supplementary figure 4d).

The general effect of RecA is to inhibit the 5-3 unwinding activity of RadA that loads onto the 5'-terminated overhang. The increase in the F1a or F2a products (the RecA activation) is rather modest (particularly if errors were to be shown) compared to the inactivation of the unwinding that produces the S1 or F2b products.

The effect of RecA on RadA unwinding activity assayed on 3-J substrates is twofold. The first is to inhibit the unwinding activity of RadA in the '3' direction' of the 3-J. This inhibitory effect of RecA on RadA is 3 to 2 fold on 3J-1 and 3J-2 at the highest RecA concentration tested (Figure 5f and 5g respectively). This inhibitory effect of RecA is also effective on a 5' tailed duplex (Supplementary Figure 6e). Therefore, it can be explained by the binding of RecA on ssDNA that impedes the loading and subsequent 5' to 3' translocation of RadA on the same ssDNA strand. The conclusion of the experiment performed with the RecAK85A mutant shown in Figure 5h supports this conclusion. In this experiment, the inhibitory effect of RecA on RadA is no longer seen, even if this point mutant of RecA is still able to interact with RadA in the BacTH assay. The second effect of RecA on the unwinding activity of RadA assayed on 3-J substrates is to

promote the unwinding of the 3-J substrate in the '5' direction'. The absolute level of F1a and F2a products obtained is indeed modest, especially in the experiment performed with RadA alone, but the increase in these products at the first RecA concentration tested is significant (more than 10 and 6 fold; Figures 5f,i and 5g, respectively). At higher RecA concentrations, a lower stimulation is observed. Again, this stimulatory effect of RecA is not seen with the RecAK85A, indicating that RecA has to interact with the ssDNA arms of the 3J molecules to promote their unwinding in the 5' direction in the presence of RadA.

The results in h/i are not quantified and would be better repeated on 3-J2 which produces more measurable product.

The results in h/i have been quantified and the figure has been changed accordingly. The 3-J1 substrate has been used because it generates more products in the 5' direction upon incubation with RecA and RadA, which was therefore more appropriate to test the RecAK85A and RadAC27A mutants and probe their functional interplay.

It is not clear on the 3-J1 and 3-J2 substrates why the loading would be on the "bottom" strand and go in the "5' direction" (these terms are confusing –see below). Why wouldn't some loading events be directed to the "top strand" and go in the "5' direction"? The 3-J substrates could be said to mimic both junctions (as shown in Figure 5b). It could be that this IS is happening but this result is hidden by the background of helicase activity.

There is a key point raised here about the unwinding of the 3-J substrates in the '5' direction'. We reveal that RadA is structurally and functionally related to DnaB-type helicases. A feature common to these hexameric motors is to be loaded and to translocate along ssDNA to unwind duplex DNA in their path in the 5' to 3' direction. For this reason (that has been proven to be the case for RadA in Figure 4), we propose that the unwinding of the 3-J substrates in the 5' direction observed in the presence of RecA and RadA reflects the loading and translocation (in the 5' to 3' direction) of RadA on the bottom strand of the 3-J molecule. Importantly, in this proposal, RecA allows RadA to access the bottom strand of the 3-J. Indeed, another possibility would be that RecA would permit the loading of RadA on the top strand. However, if it was the case, RecA would also reverse the translocation polarity of RadA in order to generate the F1-a and F2-a product from the 3J-1 and 3J-2 substrates. To test this hypothesis, we performed a helicase assay with 3' tailed DNA duplex in the presence of RecA and RadA. The results have been added in the revised version of the manuscript in Supplementary Figure 6e (left panel) and showed no unwinding of the 3' tailed duplex upon co-incubation RadA with RecA. This new result further supports the model that RecA mediates the loading of RadA on the bottom strand of the 3-J substrate to promote its unwinding in the 3' direction. This experiment is presented as follows in the results section of the revised manuscript, in the section entitled **"Interplay between RecA and RadA at three-way DNA junctions":**

The same experiment reproduced on a reciprocal 3' tailed duplex did not lead to its unwinding (Supplemental figure 6e), showing that RecA did not act on RadA to reverse its translocation and unwinding polarity. Therefore, this experiment further supports the notion that RecA promotes RadA loading on the bottom strand of the 3-J substrates to promote their unwinding in the 3' direction.

The authors need to design different substrates/more sophisticated assays to fully explore the model.

On the basis of our new results showing that RadA is a functional ring-shaped DnaB-type helicase and revealing its interplay with RecA on 3-J substrates mimicking the boundaries of a D-loop, it is clear that an important perspective is to further explore the working model depicted in Figure 6. In particular, it is important to use other substrates and other assays to clearly understand the interaction between RadA and RecA. This could be achieved through single-molecule assays. This represents however a complete and challenging study largely beyond the scope of this work.

The authors make a lot of effort to make the paper understandable. However, it is still confusing to follow and in particular Figure 5 switches between discussion of 5-3 helicase activity and 5' direction unwinding, which is still 5-3 helicase activity.

We agree that the helicase assays could be easier to follow. After discussion with several colleagues, the consensus was that this was the clearest way to present the data. We would gladly accept concrete suggestions from the referee on how to make them clearer for the broader audience to which this paper is aimed. Otherwise we will maintain the nomenclature of the '5' and 3' directions of unwinding from the D-loop as the polarity of the invading ssDNA in the text.

Reviewers' Comments:

Reviewer #1:

Remarks to the Author:

The authors have satisfactorily addressed all concerns raised from the previous review. Publication in Nature Communications is recommended.

Reviewer #2:

Remarks to the Author:

The authors have satisfactorily responded to my comments and made appropriate changes/clarifications to the paper.